# Tipping cascades between conflict and cooperation in climate change

Jürgen Scheffran[1], Weisi Guo[2], Florian Krampe[3], Uche Okpara[4]

[1]Institute of Geography, Universität Hamburg, 20144 Hamburg, Germany
[2]Centre for Autonomous & Cyberphysical Systems, SATM, Cranfield University, Bedford, MK43 0AL, UK
[3]Stockholm International Peace Research Institute (SIPRI), Solna, 169 72, Sweden
[4]Natural Resources Institute, University of Greenwich, Medway Campus, Chatham Maritime ME4 4TB , Kent, UK

*Correspondence to*: Jürgen Scheffran (juergen.scheffran@uni-hamburg.de)

**Abstract**: Following empirical research on the dynamics of conflict and cooperation under climate change, we discuss complex interactions and transitions, connected to models of tipping points, compounding and cascading risks. In the context of multiple crises, pathways in the climate-security nexus are analysed, with conditions of and societal responses to conflict risk and climate vulnerability. System and agent models of conflict and cooperation are considered to analyze dynamic trajectories, equilibria, stability, chaos and empirical simulations as well as adaptive decision rules in multi-agent interaction and related tipping, cascading, networking and transformation processes. A bi-stable tipping model is applied to study transitions between conflict and cooperation, depending on internal and external factors as well as multi-layered networks of agents, showing how negative forces can reduce resilience and induce collapse to violent conflict. The case study of Lake Chad is used to demonstrate climate change as a risk multiplier in the model. For poor governance, community behaviour is facing low barriers to climate stress which can tilt towards conflict, while resilience can build barriers against it. Narratives confirm that forced migration and militant forces lower the barrier and the chance for cooperation. Adaptive and anticipative governance based on integrative research and agency can prevent and contain climate-induced tipping to violent conflict and induce positive tipping towards cooperative solutions and synergies, e.g. through civil conflict transformation, environmental peacebuilding and forward-looking policies for Earth system stability.

**Keywords:** Tipping cascades, climate-conflict nexus, conflict and cooperation, conflict models, complex networks, agency.

## 1. Introduction: Complexity challenges in security, conflict, and multiple crises

In the Post-Cold War era, the international security landscape has become increasingly complex, expressed in the "complexity turn" of international relations (Urry, 2005; Scheffran, 2008). In the new world of disorder and multiple crises cascading chains of events are emerging, including complex social interactions and self-reinforcing collective dynamics such as stock market crashes, migration, pandemics and violent conflict that increasingly challenge international stability. A particular form of social instability contributing to crises is conflict between incompatible values, priorities, and actions of agents who undermine each other's values and provoke hostile responses, leading to escalating interactions in situations where conflicts are not resolved. While there is substantial understanding regarding the dynamics of conflict and cooperation based on quantitative and qualitative empirical methods, systems dynamics and modeling approaches have played only a marginal role in the analysis of transitions between conflict and cooperation and related switching of behaviour. Micro-macro transformations in international security and multi-scale self-organisation may accelerate beyond tipping points. In an interconnected world at the edge of chaos (Kavalski, 2015), changes in one part of the world can have significant impacts elsewhere and propagate through systemic networks like a domino effect or chain reaction with multiplying consequences.

A better scientific understanding of the underlying complex interactions is a prerequisite to stabilize the Earth system and to enable forward-looking adaptation-oriented policies that prevent violent conflict and enable cooperative stabilization of the

Earth system. This is one of the first studies connecting research on conflict and cooperation in climate and environmental change, with research on tipping points, compounding and cascading risks. While the first field has been addressed mostly by quantitative statistical analysis of large-scale data or case-based qualitative research, the second field is rooted in conceptual and modeling approaches of tipping and cascading events, including system and agent-based models. Aiming for an interdisciplinary approach bridging perspectives and methodologies of natural and social sciences in both fields is challenging and promising at the same time. In a world of multiple crises aggregating into a polycrisis one discipline alone cannot address the complexity of interconnected security issues (Scheffran, 2016; Homer-Dixon et al., 2022; Lawrence et al., 2024). Addressing research questions on conditions for stability and instability across the conflict-cooperation spectrum, and the role of adaptive and anticipative governance, we aim to demonstrate the relevance of tipping cascades from different perspectives.

After the introductory part on complexity challenges in today's crisis landscapes, Sect. 2 provides key terms related to compound risks, tipping points and cascades, as well as conflict and cooperation in socio ecological systems with exemplary cases. Sect. 3 draws on a selective overview of the literature on climate change as a risk multiplier, the environment-conflict nexus and pathways connecting climate-related vulnerability, violence, and tipping cascades between conflict and cooperation for selective examples. A focused review of models applicable to conflict and cooperation and related tipping dynamics is given in Sect. 4, preparing the design of a bi-stable model framework in Sect. 5 for integrating tipping processes in conflict-cooperation studies. Model applications in climate conflict analysis are exemplified for the case study of the Lake Chad hot spot in Sect. 6. Challenges of governance and management of negative and positive tipping in conflict and cooperation are discussed in Sect. 7, including conflict transformation and environmental peacebuilding, followed by a summary, discussion and conclusions. Our study goes beyond a review of the research literature by merging complementary research streams and presenting potential pathways towards future research.

## 2. Compound risks, tipping points, and cascades in social systems

The growing research on compound events, tipping elements, chain reactions and risk cascades provides insights on complex transitions between qualitatively different states in natural and social systems, which accordingly are adequate to learn about interactions and transitions between conflict and cooperation.

### 2.1 Compound events

Environmental risks can be amplified by the combination of multiple stressors and hazards, the co-occurrence of which contributes to societal and/or ecological risks across temporal and spatial scales. Compound weather/climate events are defined "as the combination of multiple drivers and/or hazards that contributes to societal or environmental risk" (Zscheischler et al., 2018; Zscheischler and Raymond, 2022; Pescaroli et al., 2024). Risks of hazards are a function of exposure, vulnerability, and adaptive capacity of affected systems and populations which can interact in complex ways. A key risk factor is weather which comprises short-term atmospheric phenomena (e.g. rainfall over several days) and extreme events (e.g. storm, flood, and heat of certain intensity or duration), while climate refers to long-term conditions reflected in the mean and variance of data over decades (Dahm et al., 2023). There are numerous examples in fragile countries where exposure to weather-related hazards affects societies with high vulnerability and low adaptive capacity, triggering compounding disasters (e.g. Indus floods in Pakistan 2022 or tropical storms in Mozambique). They can also overwhelm adaptive capacity in wealthy countries, such as hurricanes in the United States where heavy rainfall and storm surge compound in devastating damage in urban centers (e.g. Katrina in 2005, Sandy in 2012, Harvey and Irma in 2017). Heavy snowfall in parts of Germany in November 2005 caused

power outages for some 250,000 people for several days, and a snowstorm in North America 2013/2014 major power cuts for hundreds of thousands of people, leading to partial failure of communication and transport systems (Scheffran, 2016). Major floods in Western Germany in July 2021 demonstrated that unpreparedness can leave more than hundred casualties, cause billions of Euros in damage and devastate the infrastructure. When short-term weather shocks occur more frequently and intensely with climate change, they can turn into a long-term force threatening ecological and social systems that cannot adapt, destabilizing ecosystems, damaging infrastructures and coastal protection, and provoking behavioural changes when living and working conditions become unbearable. They turn into large-scale events such as the vegetation fires in the summer of 2010 in Russia with severe air pollution and impacts on crops and human health (Reichstein et al. 2021), affecting global food markets and social stability with conflicts contributing to the compound of problems. Tradeoffs and synergies of compound effects are represented in the nexus approach, such as the water-food-energy nexus (Rasul, 2015); Albrecht et al. 2018) or the climate-conflict-migration nexus (Brzoska and Fröhlich, 2016; Watson et al., 2022).

## 2.2 Tipping points and thresholds

A well-known phenomenon from chaos theory is the sensitivity to initial conditions, symbolized by the butterfly effect when small changes can have large effects which at critical thresholds to instability and bifurcation (Scheffer 2009) may trigger or prevent a phase transition into new states (such as gas, liquid and solid phases of matter) that do not have to be self-enforcing and irreversible. A related concept is tipping point which according to Milkoreit et al. (2018: 9) is a "point or threshold at which small quantitative changes in the system trigger a non-linear change process that is driven by system-internal feedback mechanisms and inevitably leads to a qualitatively different state of the system, which is often irreversible." Most prominent are tipping elements in the climate system which include self-reinforcing melting of the Greenland and West Antarctic ice sheets, release of frozen greenhouse gases such as methane, weakening of the North Atlantic Current, or changes in the Asian monsoon (Lenton et al., 2008; 2023). Above a critical temperature threshold, amplification effects and chains of events could lead to fundamental Earth system changes reaching planetary boundaries (Steffen et al., 2018). With the broad definition of tipping points they cannot only be found in natural systems but also in social systems (Otto et al., 2020; Franzke et al., 2022). For tipping points in political contexts, it has been suggested "that events and phenomena are contagious, that little causes can have big effects, and that changes can happen in a non-linear way but dramatically at a moment when the system switches" (Urry, 2002:8). Differences of social tipping are critically emphasized, such as agency, complexity, non-reductionist or non-deterministic mechanisms which can apply to both negative and positive tipping processes, depending on the evaluation of advantages and disadvantages. (for critical perspectives see Milkoreit, 2023).

## 2.3 Cascades and chain reactions

Tipping points may trigger more tipping points, leading to "tipping cascades", including domino effects and chain reactions (AghaKouchak et al., 2018; Klose et al., 2021; Lenton, et al., 2023). While a tipping point usually refers to exceeding a threshold (the first domino falling), tipping cascade represents the chain sequence of following events (more dominos falling). The difference is not always clear or easy to distinguish because it depends on case-specific circumstances, including the couplings of system variables and agent responses (the length, density, number of dominos and possible blocking interventions). Individual systems or communities can have a tipping point and threshold at which it tips, but as the effects of tipping variables are inducing changes in other variables the question is how far the chain continues and is spreading through the network of connections and sensitivities, extending a single tipping into a tipping cascade until a new stable state is reached. Since transient behaviour is hard to predict and control, we cannot simply say whether the whole system tips or only parts of

it. How far the spreading of dominos continues, when it stops or is recovered, depends on the heterogeneity and connectivity in space, time and context (for more on definitions see Lenton et al., 2023; Kopp et al., 2016).

An example from biology and health is the Corona pandemic, in which all humans are part of a spreading virus infection with tipping and cascading mechanisms beyond a critical infection rate and a decay below. A physical example is the exponential chain reaction of nuclear fission beyond critical mass or density, which is uncontrolled in the atomic bomb and held at the threshold of criticality in the nuclear reactor by control rods to extract energy. When control is lost, a reactor accident can set in motion local and global impact chains, as demonstrated by the nuclear disasters at Chernobyl (26.04.1986) following an

accident, and in Fukushima (11.03.2011) when a tsunami flooded parts of the Japanese coast, claimed many lives, and triggered explosions in several nuclear reactors, spreading radioactivity globally through the atmosphere and the ocean. The consequences affected the Japanese power grid, the nuclear industry, stock markets, oil prices and the global economy when automobile manufacturers and electronics companies cut back production because important components were not delivered from Japan. The shock waves demonstrated how a compound event can set into motion a global tipping cascade, changing the

economic and political environment, for instance triggering the energy transition in Germany (Kominek and Scheffran, 2012).

Revolutions in history were often associated with loss of control by the existing order and following tipping cascades, such as the French revolution following the storming of the Bastille on July 14, 1789 which destabilized the order of Absolutism. Two hundred year later, the fall of the Berlin Wall on November 9, 1989 triggered a cascade of dissolving political regimes in

Eastern Europe which marked the collapse of the Soviet world order, German unification and the end of the Cold War, induced by Mikhail Gorbachev's failed attempt to reform the Soviet Union which lost control. This turning point in world history opened the following complex era of crisis and transformation toward a globalized international system that continues to be unstable (Jathe and Scheffran, 1995; Scheffran, 2008). A tipping cascade also evolved from the financial crisis reaching a climax with the bankruptcy of Lehman Brothers on September 15, 2008, and a subsequent international banking crisis, powered

by dubious speculations, lending practices of financial institutions and short-sighted human behaviour, escalating responses and self-reinforcing interaction between rating agencies and government measures. This created an explosive situation, pushing the global financial system to the brink of collapse (Barrell and Davis, 2020). Production losses, bankruptcy of companies or stock market crashes propagated across global networks and markets, diverting hundreds of billions of dollars of state funding for stabilization. In Europe, the global economic crisis was followed by a crisis in southern Europe, particularly in Greece.


## 2.4 Tipping cascades in conflict and cooperation

Conflict and cooperation are important forms of social interaction. Conflict generally refers to social or political incompatibility of interests, values or actions between social actors who fail to reduce their differences and tensions to tolerable

levels, escalating the conflict by continued actions, including protest, resistance and violent acts causing mutual losses. Cooperation is the opposite interaction when the interests, values or actions of social actors are not only compatible but even beneficial to others, leading to mutual gains that stabilize this interaction (Scheffran and Hannon, 2007). Both conflict and cooperation are affected by human motivations and values (e.g., life, health, income, assets) as well as by capabilities and opportunities (e.g., money, resources, vehicles, equipment, technology). They have direct and indirect impacts on human

actions and responses that can drive and inhibit conflictive and cooperative actions, destabilizing interaction to a downward spiral of violence and a vicious circle of conflict escalation (Buhaug and von Uexkull, 2021) or be stabilized to a virtuous circle of solutions by governance mechanisms and institutional policies, separated by negative and positive tipping points when interaction is qualitatively changing (Lenton et al., 2023, Chapt. 2.4 and 4). Near thresholds of instability, a seemingly minor change can trigger rapid transitions between conflict and cooperation, escalation and de-escalation, war and peace, if no

mediating and stabilizing measures are taken. Compound events, tipping elements and risk cascades can combine in tipping cascades triggering the transition processes between conflict and cooperation.

Major violent events are often related to cascades marked by initiating dates such as World War 1 (28.07.1914), World War 2 (01.09.1939), terror attacks on the world trade center (11.09.2001), Russia-Ukraine war (22.04.2022) or Hamas-Israel war

in Gaza (07.10.2023), each of which was pre-empted by a sequence of events building up to decisions launching violent acts and followed by the consequences and responses. Combinations of compounding and cascading dynamics do not necessarily require a particular tipping date but can include a sequence of events over longer time periods which occur in many armed conflicts around the world. What they have in common is that the affected world after conflict tipping is perceived as different than before, which does not exclude that the dynamics can be influenced towards escalation and more violence or to

deescalation ending it, pointing to the relevance of agency preventing or reversing the mechanisms of violence.

## 3. Crises, conflict and cooperation in climate and environmental change

### 3.1 Climate change as a crisis multiplier

Rising global temperature above a certain threshold may exceed the adaptive capacity and resilience of natural and social systems, trigger tipping cascades and spread through networks of connections, including disasters and weather extremes, famines and epidemics, poverty and refugees, crimes and riots, violent conflict and terrorism. There can be drastic changes in

individual and collective action, or institutional settings and governance, legal and economic arrangements, and long-term effects on social norms and values. Tipping in natural systems can interfere with social tipping dynamics in negative and positive ways, and trigger tipping cascades across multiple systems, with asymmetric response mechanisms across social systems (e.g., economic, health, social cohesion) and scales (household, regional, country to global levels). Cascading stressors and risks by sudden- and slow-onset climate-related events affect non-linear behaviour and feedback within adaptation limits.


Climate change can interact and multiply with other risks and crises potentially leading to a downward spiral, where abrupt and extensive climatic changes and extreme events could cascade through the globalized economy (Stern, 2006; Onischka ,2009), potentially leading to economic shocks, stock market crashes, loss of production, supply shortages and price increases. Climate-related extreme events can spread through global supply chains and societies (Levermann, 2014), triggering cascades

in social networks, in protest movements, elections, revolutions, mass migrations or violent conflicts (Kominek and Scheffran, 2012). Sometimes switching results from triggering events or social movements with self-enforcing cascading sequences, e.g., when an action taken by one actor provokes more intense actions by other actors. The key question is whether climate effects and related climate shocks are strong enough to result in tipping (Kopp et al., 2016) which is not generally the case but dependent on specific circumstances such as resilience, cohesion and mutual support between communities. Some societies

may have low, others high barriers to tipping, e.g. from societal organisation or mutual support (see Sect. 5).

A case which has been extensively discussed is the Arabic Spring of 2011, a series of social and political protests and upheavals that provoked regime change in several countries in the Middle East and North Africa (MENA). The self-immolation of Mohamed Bouazizi in Tunisia on December 17, 2010 became a tipping point of uprisings spreading to Libya, Egypt, Syria

and Yemen, multiplied and accelerated by the Internet and social media (Kominek and Scheffran, 2012), which enabled and motivated others to join the protest movement. Facing repression from the regime, the self-organized resistance remained largely peaceful but the situation turned violent, especially in Libya and Syria. Some sources suggested a link with the sharp

rise in food prices at the turn of 2010–2011 and that extreme weather events might have contributed to these processes (Johnstone and Mazo, 2011), such as drought in Russia and China 2010 and 2011, which exerted pressure on the international market price of wheat and influenced the availability of food products. This coincided with other factors that increased food prices, including high oil prices, bioenergy development, and speculation on food markets. The consequences affected much of MENA with large food importer where low incomes and high food spending affected food security. The sharp rise in bread prices magnified the existing public dissatisfaction with the governments and triggered political protests. While no relevant protests took place in Israel or the Gulf States, in Tunisia, Libya and Egypt governments were overthrown, while in Syria and Yemen civil wars emerged, each with a cascade of consequences from refugee movements to terrorism reaching Europe. In this complex pattern of overlapping stressors, climate change was not the main cause, but contributed as an additional stressor overwhelming government control leading to tipping cascades. The political upheavals until today affect the stability of the Mediterranean region but have also induced cooperative mechanisms of energy and climate governance.

## 3.2 The environment-conflict nexus

Environmental change is potentially associated with a wide range of conflictive issues. The concept of security has been expanded to include ecological dimensions and the availability of natural resources. Environmental conflicts concern the use and degradation of exhaustible and renewable resources, regenerated in metabolic cycles, depending on the functioning and stability of ecosystems, which in turn are affected by conflicts. A lacking balance between human demands and available resources, together with an insufficient use and inequitable distribution of resource benefits and risks, contains a significant conflict potential. Competition, grievance, or greed can arise from scarcity and/or abundance of resources (Okpara et al., 2016), including situations of differing interests, values, incentives, and priorities amongst resource users.

Environmental change and violent conflict together can weaken social relations and social capital, e.g., when weather extremes disrupt infrastructures and stability of society, or violent conflict constrains the capacity of people and countries to adapt to climate change, which in turn makes recovery and peacebuilding more difficult (Juhola et al. 2022, Krampe, 2019). This can weaken the resilience of communities and institutions in places like Iraq and Somalia, hindering their ability to maintain peace. Conversely, conflict-related effects such as displacement or disruption of livelihood practices may impede the capacity of communities and institutions to adapt to climate change in places like Afghanistan or Mali. Having lost savings and assets in conflict, impoverished communities in low-income countries are highly vulnerable to future risks and have few resources to respond: "Vulnerability is higher in locations with poverty, governance challenges and limited access to basic services and resources, violent conflict and high levels of climate-sensitive livelihoods (e.g., smallholder farmers, pastoralists, fishing communities) (high confidence)." (IPCC, 2022). The double exposure to environmental and conflict risk is associated with compound effects where "environmental change can make societies more vulnerable to violence which in turn can make societies more vulnerable to environmental change, leading to a trap from which escape is difficult" (Scheffran et al., 2014: 375). It is difficult to separate mutually enforcing vulnerabilities to climate and conflict that escalate in a spiral of violence and amplify cascading crisis events beyond critical thresholds connected through tele-coupling (Franzke et al., 2022).

## 3.3 Pathways of climate-security interaction

First, we consider climate change as a conflictive issue, from disputes over scientific predictions, impacts und uncertainties of climate change to violent conflicts fuelled by the security risks of climate change or measures to prevent and address climate risks. Studies on climate-conflict linkages discuss the effects of various climate phenomena (e.g., change of temperature and precipitation, resource availability, weather extremes, sea-level change) on different phases of conflict (onset, initiation,

escalation, prolongation, termination, prevention) or different types of conflict (e.g., communal, rebel, farmer-herder conflicts). Conflict parties can be nations, individuals, parties, companies, trade unions, activist groups and generations, among others.

Understanding the relationship between climate change and security risks has advanced significantly in recent years (Uexkull and Buhaug, 2021). While there were differing interpretations in past IPCC reports, research generally agrees that climate change not only exacerbates the causes and effects of conflict, but also affects the ability of communities and institutions to move towards cooperation and establish and maintain peace in specific contexts (Gleditsch and Nordås, 2014). The latest IPCC summary reaffirms with high confidence: "Climatic and non-climatic risks will increasingly interact, creating compound and cascading risks that are more complex and difficult to manage" (IPCC, 2023). A substantial body of qualitative and quantitative studies from various disciplines provide new insights into the context, timing, and spatial distribution of climate-conflict risks (De Juan, 2015; Brzoska and Fröhlich, 2016; Buhaug, 2015; Abrahams and Carr, 2017; Scheffran, 2020a; Rodriguez et al., 2019). Climate change is not the sole cause (Mach et al., 2019; Sakaguchi et al., 2017; Scartozzi 2020; Ge et al. 2022), but can undermine human livelihoods and security by increasing vulnerabilities, grievances, and political tensions through indirect and sometimes non-linear pathways, resulting in human insecurity and violent conflict risks (van Baalen and Mobjörk, 2017; Koubi, 2019; Uexkull and Buhaug, 2021). The main purpose is not to prove a general and significant impact of climate change on conflict but to understand the sensitivities connecting them in both directions and the role of cooperation as a possible response mechanisms mitigating climate conflict which supports complex transitions and tipping.

Research has identified five risk dynamics that illustrate the complexity of climate-related security risks (SIPRI 2022):

1. Compound risks, in which the simultaneous interaction of two or more risk factors results in a greater risk complex, as the factors mutually reinforce each other.
2. Cascading risks, in which an event creates a risk that leads to subsequent, sequential risks, generating an increasingly escalating risk potential like a snowball effect.
3. Emergent risks, in which two or more temporally and spatially independent factors create new risks that would not have existed without the previous ones.
4. Systemic risks, in which multiple risk factors interact in such a way that they cumulatively threaten a societal and/or ecosystem in parts or as a whole.
5. Existential risks, whose impacts are so severe that they threaten the existence of a country or culture, for example.

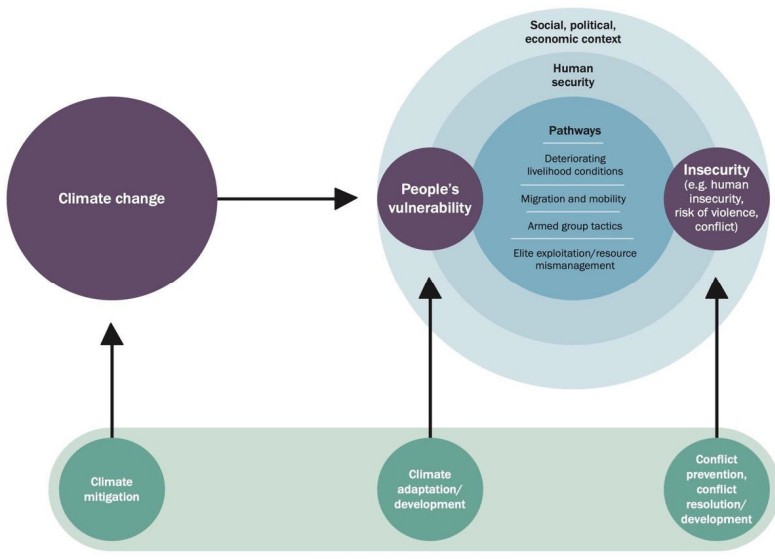

**Figure 1: Pathways and entry points of climate-security interaction (source: SIPRI, 2022: 63)**

These pathways indicate the complexity of climate-related impacts on societies which reduce the system to its core elements "climate change", "people's vulnerability" and "insecurity" (SIPRI, 2022) connected through four climate-security pathways, translating climate-related vulnerability into physical violence (Fig. 1): (1) livelihood deterioration, (2) migration and mobility, (3) existence of tactical opportunities for militant and armed actors, and (4) elite exploitation and political and economic grievances (Mobjörk et al., 2020).

1. Climate change undermines the livelihoods of societies, potentially increasing the risk of conflicts. For example, changing weather patterns significantly impact agriculture and livestock farming, putting pressure on societies or specific populations whose income depends primarily on agriculture. This in turn may lead to tensions and violent conflicts between different groups, particularly at the local level. In Somalia, for instance, the reduced resilience of the impoverished population affected by prolonged conflicts forces people to engage in informal activities such as illegal logging for charcoal production to secure their survival (Shaik Dahir, 2023). The loss of livelihoods, which are often an integral part of personal identity, also leads people to join extremist groups not due to ideological conviction, but rather due to personal needs.

2. Climate impacts and conflict risks may result in displacement and changing mobility patterns, which can interact in complex ways with multiple and context-dependent outcomes, sparking divergent viewpoints and controversies precluding simple threat perceptions (Issa et al., 2023). Counterproductive responses extend security policy to fight symptoms and not causes, discouraging migration and stigmatising displaced populations, such as tightening border controls. Migration decisions of exposed populations depend on personal and social circumstances (Koubi et al., 2022). There is a tradeoff between motivating drivers of migration and diminishing capability to move, leading to involuntary "trapped" populations (Benveniste et al., 2022). While migration under climate and conflict may entail exposure to new risks, it can reduce some risks and serve as an adaptation and risk management strategy (e.g. Scheffran et al., 2012c; Gioli et al., 2016; Adger et al., 2024), also raising critical questions (Vinke, 2020). For instance, floods, droughts and deteriorating living conditions may drive people to urban or rural areas with limited economic opportunities, further straining local resources and causing tensions. On the other hand, adaptation can reduce incentives to move, especially in dryland regions with large seasonal and annual variations in environmental conditions. When nomadic populations move into new regions bordering their traditional routes due to seasonal shifts and their herds graze on land cultivated by sedentary farmers, this may disrupt traditional mechanisms that have regulated the coexistence of these groups in the past (Bukari et al., 2018). Unstable power relations or armed conflicts can quickly escalate. In many cases environmental impacts are linked to temporary, short-term, or domestic migration (Hoffmann et al., 2021). Future research can develop holistic perspectives and synergies of management strategies (Simpson et al., 2024).

3. Climate change influences the behaviour of armed actors and directly affects conflict dynamics. This includes not only the military readiness of state and non-state actors, but also changing power dynamics. For example, insurgent groups in Mali and Somalia may find it easier to move in flooded areas compared to conventional forces, making the latter more vulnerable. Additionally, extremist groups can exploit societal grievances over the state's handling of climate change impacts to further their agenda, such as recruiting members or mobilizing support.

4. Climate-related security risks also affect governance structures and can exacerbate existing governance challenges. Climate impacts can strain government capacity and resources, leading to weakened governance institutions and ineffective policies. This can result in social unrest, loss of trust in institutions, and erosion of state legitimacy. In addition, climate change can create new power dynamics, with some actors gaining or losing influence because of changing resource availability or shifts in geopolitical interests.


Interacting pathways and tipping cascades can create highly complex climate-conflict relations difficult to contain or control in time, space and extent. An integrated framework combines various perspectives – political and economic, social, human and psychologic, governance and institutions, social-ecological and environmental security. Multicausality of conflict and cooperation and feedback to climate complicates the picture. For instance, when conflicts escalate, exhibiting a tipping

dynamic, they can in turn impact the Earth System environment, as warfare itself is producing excessive greenhouse gas emissions (Vogler et al., 2023), which is the case for Russia's war in Ukraine (Flamm and Kroll, 2024).

## 3.4 Hot Spots of climate-conflict pathways and tipping cascades

Climate–security pathways have been studied in hot spots around the world, some of which serve as exemplary cases for complex social interactions and tipping cascades, dependent on regionally specific conditions where pathways overlap and mechanisms may induce or prevent tipping in conflict (Sect. 6). Much focus is on the Lake Chad region where climate extremes interact with water and food security, resource exploitation and arms transfers, self-perpetuating cycles of violent conflicts, displacement and terrorism, placing the region on the edge of systemic criticality and conflict tipping. To overcome the

sampling bias towards the well-studied African continent (called streetlight effect) (Adams et al., 2018), more attention should be on other regions. The Syrian civil war following the uprising in 2011 has been considered as a striking case for tipping points and cascades of security risks, conflicts and refugees affecting the stability of the Mediterranean. South Asia is particularly sensitive to climate change and its extremes, interacting with low development, high population density, dependence on vulnerable agriculture, and armed conflicts in complex ways, which are subject to participatory engagement,

risk management and adaptation by farmers, households and institutions to secure human livelihoods and prevent tipping points (see Sect. 6.2).

## 3.5 Environmental cooperation, positive tipping, and agency

Despite the ability of human agency to adapt, social tipping points pose challenges to established governance arrangements, social institutions and norms, identities and worldviews. When overstepping adaptation limits agents may no longer be able to avoid intolerable risk (Dow et al., 2013). However, risk dynamics and conflict tipping mechanisms are not deterministic but influenced by context, conditions and agency. New adaptation practices go beyond incremental adjustments avoiding maladaptation and toward transformational adaptation (Juhola et al., 2022). In changing environmental and climate conditions

not only conflict but also cooperation is a possible response – e.g., when governments and societies collaborate and build alliances around environmental challenges, or initiate agreements and policy frameworks, leading to shared goals, fostering trust building and social cohesion (Lejano and Ingram, 2009; Huitema and Meijerink, 2018). Mutual adaptation of actions or institutional control mechanisms can stabilize the interaction, contain conflict or contribute to environmental peacebuilding. Mixed cases are possible, e.g. when violent conflicts do not preclude cooperation and co-existence between conflict parties

(Bukari et al., 2018). Influencing the transitions between conflict and cooperation (how they emerge, co-exist and shape social systems) is critical for defining pathways of transformation to peace and human or planetary security, including positive tipping cascades (Lenton et al., 2023) and how they can shape (and be shaped by) human responses, climate policies and negotiations across scales.

Considering the global nature of climate change, cooperation among nations is essential to drive effective climate policies. International climate agreements, such as the UN Framework Convention on Climate Change (UNFCCC) and the Paris Agreement, are examples of cooperative governance at global level (Bodansky, 2016). Such agreements provide a framework

for countries to work together to set beneficial adaptation, mitigation and technology transfer goals, and build solidarity and cooperation (e.g., support most climate vulnerable people and countries). Cooperation also happens at regional, national, and local levels, where stakeholders collaborate to develop and implement climate policies (Adger and Jordan, 2009). This may involve partnerships among governments, businesses, and civil society to foster innovation, promote renewable energy or implement climate adaptation (Pattberg and Stripple, 2008). Cooperative actions such as knowledge-sharing, capacity-building, and collaborative governance can help to create synergy and facilitate effective climate policy implementation.

The situation is complicated by human and political responses including policies to prevent and address climate impacts which can lead to tensions over mitigation and adaptation, disaster management and damage limitation, climate geoengineering, the (un)fair distribution of costs, risks, and benefits of climate change or protests against inadequate or insufficient climate action (Scheffran and Cannaday, 2013) which requires climate policies that are conflict sensitive (Nadiruzzaman et al., 2022). With increasing warming, these dynamics might become more conflictive and be linked to different national interests, priorities, and values (Victor, 2011). Engagements to mobilize, allocate and distribute climate finance can lead to conflict, also distribution of responsibilities and burdens on net zero transitions. For example, countries have divergent views on the level of climate mitigation actions required, as well as the allocation of costs and sharing of benefits. Disagreements over mitigation targets, finance mechanisms, and technology transfer can block international climate negotiations since countries have different levels of capacity, vulnerability, and historical responsibility. Within countries, conflict can arise between different economic sectors, e.g., between fossil fuel-intensive industries and renewable energy proponents, or between environmental advocacy and unsustainable economic gains. In conflict zones, conflict economies thriving on the extraction of mineral/environmental resources can hinder the adoption of climate policies, leading to policy gridlocks or delays in decision-making.

While positive tipping cascades have been explored in fields of energy, food, transportation or finance (Eker et al., 2023), they are also potentially relevant in research on environmental cooperation. First approaches have been discussed qualitatively for possible transitions from cycles of violence to cycles of cooperation, e.g. in Kenya and Sudan (Scheffran et al., 2014), or for migration networks for climate adaptation and co-development in renewable energy and water in Northwest Africa (Scheffran et al., 2012c). They may be particularly relevant in the growing field of environmental peacebuilding to understand conditions when they become leverage mechanisms for preventing climate conflict. Examples on the local scale are the Ecopeace Good Water Neighbors project between Israel, Palestine and Jordan; collective rice production in Nepal; shrimp farming, mangrove fishery and flood protection in Bangladesh, among others (Schilling et al., 2017; Ide, 2019). On larger scales promising is North-South collaboration in decentralized renewable energy projects in rural areas of Africa or South Asia without power grids, for economic development, local value chains, internet, mobile communication, education and health, better living conditions, jobs, and poverty reduction which can be combined in positive compounding, tipping and cascading success stories (Sect. 7).

Largely neglected in this research are models that analyze tipping dynamics in climate-related conflict and cooperation within the larger framework of Earth-System Dynamics (Franzke et al., 2022). Social tipping points and cascades are shaped by cross-scale feedback in social systems and human agency (Cash et al., 2006), combining system and agent models to explore and enable the analysis of complex interactions and multi-faceted governance (Hochrainer-Stigler et al., 2020).

## 4. Models of tipping in conflict and cooperation

Modeling of tipping dynamics in conflict and cooperation can be embedded into an integrative framework of Earth-System Dynamics, connecting climate change, natural resources, human security, and societal stability (Scheffran et al., 2012a,b). Climatic changes affect the functioning of ecological systems and natural resources which stress human health, wealth and security. Human responses such as migration, conflict and cooperation can merge in compounding events spreading through the network of variables connected through sensitivities affecting systemic stability in natural and social systems, including tipping cascades. Beyond integrated assessment of these complex interactions, we focus on system and agent models of tipping between conflict and cooperation in the context of climate change (Scheffran and Hannon, 2007; Guo et al., 2018).

### 4.1 System models of conflict and cooperation

System dynamics models are used to study behaviour, equilibria and stability in ecological systems, for instance the Lotka-Volterra predator-prey equations developed a century ago (Pruitt et al., 2018) which can be combined with a logistic growth function to represent two stable states (bi-stability) (Sect. 5). Of a similar type were the differential equations applied to understand conflict at the beginning of the 20th century. Shortly before World War I, the British engineer Frederick William Lanchester (1868–1946) developed mathematical "laws of warfare" to win or lose a battle based on the number and efficiency of forces. Around the same time the British physicist, psychologist, and pacifist Lewis Fry Richardson (1881-1953) who is well known for his contributions to weather forecasting, applied mathematical modeling to better understand arms race and predict war by finding general laws of conflict dynamics among nations (for a comprehensive review see Gleditsch, 2020). Deriving linear differential equations to describe the arms build-up in World War I, Richardson projected that the arms race between major powers during the 1930s could lead to another major war (Richardson 1960a,b). The model assumes that each country increases its own armament (force) level proportionate to the armament of an opponent (weighted by a defense coefficient) and reduces it proportional to its own armament (weighted by the fatigue coefficient) plus a "grievance" term (measuring the readiness to arm). In the "balance of forces" stability is determined by the eigenvalues of the matrix of driving and dampening coefficients. A positive eigenvalue represents an exponentially growing arms race (corresponding to instability), a negative eigenvalue asymptotic stability of force levels including disarmament. This stability threshold corresponds to a tipping point between qualitatively different states of conflict escalation and de-escalation (Scheffran, 2020b), which are partly reversible as military spending is lost for other purposes, even more the losses in warfare.

Richardson's model initiated research on the armament dynamics and a debate about its applicability to real-world conflicts (for details see Smith, 2020; Scheffran, 2020b), proposing extensions to address its deficiencies. Intriligator (1975) developed decision rules for increasing or decreasing weapons to bridge the gap between desired and current levels, based on strategic considerations on the expected outcome of deterrence and war. While the linearity and simplicity of the Richardson equations represent a few paths of system behaviour (oscillations, asymptotic decay, exponential increase), its rather simple structure does not represent the complexity of reality where reactions may be disproportionate and non-linear. Accordingly extensions were suggested with time-discrete difference equations using non-linear state-response functions explored in bifurcation and chaos theory since the 1970s, to study critical phenomena, such as self-organization, multi-stability, tipping points, phase transitions, and irreversibility. The concept of chaos in armed conflict was introduced to show that simple non-linear deterministic arms race models may lead to the breakdown of predictability (Saperstein, 1984). The chaotic dynamics in a logistic bi-stable arms race model was investigated by Grossmann and Mayer-Kress, (1989) using security-drivers and upper cost limitations, distinguishing between chaotic responses and instability which contains the risk of war. While the two-player arms race of Richardson or game theory were paradigms of conflict studies during the Cold War, chaos theory became a

paradigm for the following turbulent transformation and domino effects which may be described by complex multi-factor dynamic models with decision rules responding to rapidly changing security conditions, including socio-economic and political, technological and ecological dimensions, linking security and sustainability.

## 4.2 Agent models of conflict and cooperation

Going beyond optimal decision-making models, agent-based modeling (ABM) captures diverse societal agents that can choose and adapt their decisions and actions based on motivation, capability, and behavioural rules, according to reasoning, learning, perception, and anticipation which affect the expected outcome of tipping dynamics. Individual agents can select from a range of options adequate to their preferences and priorities, e.g., following rules of optimization, satisfaction, and bounded rationality, dependent on environmental change and decisions of other agents in game-theoretic settings. While non-cooperative games search for solution concepts such as Nash equilibria or Pareto optima, cooperative game theory has been developed for players joining coalitions. Evolutionary game theory analyses the competition among populations via replica equations that select cooperative and non-cooperative strategies according to fitness (Hofbauer & Sigmund, 1998). This helps to analyze the evolution of cooperation in experimental games (Axelrod, 1984,1997), finding sequential strategies (such as tit-for-tat) according to payoffs and social context, using social learning and positive tipping to escape from social dilemmas.

Adaptive models implement rule-based behaviour of agents, including response strategies, coevolutionary rules and action-reaction patterns. Models of artificial societies use computer simulation to study complex interaction between many agents who follow stimulus-response patterns in virtual environments (Epstein and Axtell, 1997). Building on tipping in the spatial segregation models of Schelling (1971) and Sakoda (1971), ABM uses behavioural rules and simulates multi-agent patterns of interaction, which is useful in situations of uncertainty, bounded rationality, and adaptive human action, providing a better understanding on how environmental conflict and cooperation evolve in multi-agent settings (Bendor and Scheffran, 2019). Climate and resource limitations may modify the rules and interactions, triggering conflictive or cooperative behaviour changes. Ultimately agents choose behavioural rules which create social and environmental conditions affecting these rules.

Multiple agents show collective behaviour via opinion dynamics, coalition formation, social networking, norm building, and transformative policies, including pathways, transitions and tipping between conflict and cooperation (Bendor and Scheffran, 2019; Juhola et al., 2022). An example is Epstein (2002) who finds tipping points for police efforts against civil unrest and interethnic violence. ABM can simulate cascading effects in social networks, and self-reinforcing chain reactions that could e.g., increase conflicting and antisocial behaviour (Filatova et al., 2016; BenDor and Scheffran, 2019). ABM captures macro-scale phenomena from micro-scale interactions among heterogeneous adaptive and learning agents (Filatova et al., 2013) where seemingly minor events can provoke major qualitative changes in social systems, such as the end of the Cold War and the Arabic Spring. BenDor and Scheffran (2019) suggest to apply ABMs to study environmental conflict and cooperation, as well as adaptation behaviour and institutional responses to climate-conflict risks. Societal interactions can be represented by social network analysis (SNA) which visualize the dynamic switching and tipping between alternative pathways in response to changing internal and external conditions, in particular hostile and friendly behaviour. The cascading spread in social networks has been applied to the diffusion of social behaviour, technical innovations and spatial conflict, in particular in World War I (e.g., Kempe et al., 2005; Flint et al., 2009; Maoz, 2010). Rodriguez et al. (2021) combined ABM and SNA to analyse conditions for changing mobility patterns in pastoral groups

### 4.3 The VIABLE model framework: Stability and complexity in environmental conflict and cooperation

System and agent model approaches are integrated in the VIABLE model framework which stands for "Values and Investments for Agent-Based interaction and Learning for Environmental systems" (Fig. 2a) (Bendor and Scheffran, 2019). It models the dynamic action and interaction of agents who use part of their available capabilities ($K$) as investments ($C$) with priorities ($p$) to given action pathways ($A$) that change their environment ($X$). The observed impacts of actions are evaluated in each time step based on actual values ($V$) and targets values ($V^*$) where agents are satisfied. Important parameters are the sensitivity of value to environmental change ($v_x$) and the inverse sensitivity (unit cost) of environmental change to investment ($c_x$). The respective value-cost ratio $f = v_x/c_x$ of an action indicates how sensitive and efficient its value is to investment. Negative efficiencies $f$ indicate a conflicting action path where agents violate each other's values. In repeated time steps and learning cycles agents mutually adapt their capabilities, action priorities and values to the satisfaction levels, as a function of the sensitivities between agents and the environment. Within available capability limits, agents adjust their action pathways to meet their value goals according to logistic decision rules which determine multiple equilibria where agents are satisfied.

The first model application was the Cold War arms race in the 1980s, where tipping from hostile to friendly attitudes of the superpowers was simulated, showing a chaos-like transition to nuclear disarmament which was validated when Cold War ended. The VIABLE framework was also applied to other tipping problems, e.g., in fishery, land use, energy, transportation, health, migration, sustainability, climate policy and emission trading (see Scheffran and Hannon, 2007; Bendor and Scheffran, 2019). The VIABLE model allows to study transitions between conflict and cooperation, bi-stable fixed points representing the balance of investments (social equilibrium) meeting satisfaction levels. Agents can control and stabilize or destabilize the dynamic interaction by using their capabilities and changing their action priorities to achieve their target values. If action priorities are directed towards hostile relations (damaging the values of other agents), the dynamics moves towards increasing investments and conflict escalation. In a bi-stable case agents can switch to mutually beneficial cooperation with less investments. They may also have no effect on each other's satisfaction levels (neutrality) or mixed cases (Fig. 2b). Individual agents can form coalitions by pooling some of their invested capabilities and redistributing the gains (or losses), or agree on the same values and targets, thus moving from individual to collective or institutionalized action and interaction.

The type of social interaction in the VIABLE model is represented by the interaction matrix and its stability, mathematically determined again by the eigenvalues around the social equilibrium (Fig. 2b). If agents are powerful in terms of their capabilities and efficient in using them to pursue their value goals, they can withstand, compensate, or counter-act a certain level of hostility by others, keeping eigenvalues in the stabilising (negative) range and avoiding major deviations from equilibrium. If hostile actions exceed a critical threshold, a destablising escalation may occur. Stability of social interaction can be maintained if the positive (cooperative) effects of agents on each other exceed their negative (conflicting) effects which is a tipping condition. With a growing number of agents, the complexity of the interaction matrix and the number of eigenvalues increases, including those that are potentially unstable, which is known in systems theory as the "complexity-stability" tradeoff (Scheffran and Hannon, 2007; Gravel et al., 2016). In response to tipping cascades a system can break apart into simpler ones or forms more complex ones. Mutual adaptations or institutional control mechanisms can stabilize the interaction and contain conflict.

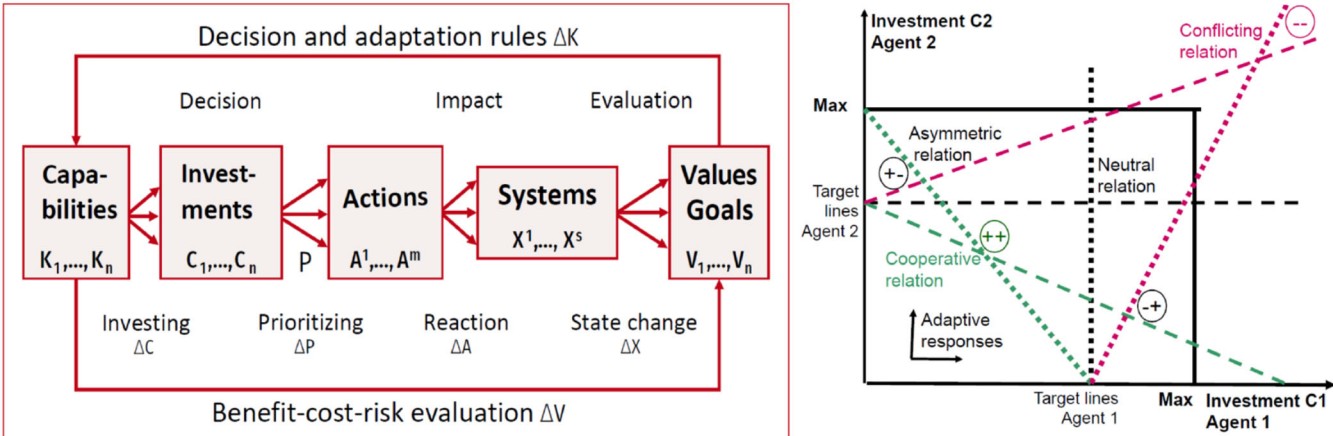

**Figure 2: (a) Dynamic VIABLE model framework for multiple agents; (b) Adaptive target lines for two agents and mutual equilibria (balance of investments) for conflicting (--), cooperative (++), mixed (+-/-+) and neutral relations.**

### 4.4 Additional models relevant for tipping in conflict and cooperation

There are various additional models for the study of tipping processes in conflict and cooperation that are specific to certain application areas, methods, and data, including nonlinear models, which we shortly referr to (Guo et al., 2023):

- *Machine learning (ML) and causal inference* identify multiple climate-conflict pathways and causal mechanisms based on large-scale data globally (Ge et al., 2022) and regionally (Xie et al., 2022). Most recent work (e.g., in social transformation) integrates bifurcation behaviour with neural networks to harmonize data driven prediction with expert-informed climate fragility indices (Sun et al., 2022).

- *Excitation-cooling* models represent specific violent conflict processes, where a successful attack leads to more attempts in the same area while excitation can be cooled down by security forces increasing patrols and preventions which can be perceived as a tipping process (Tench et al., 2016). Case studies covered data from Northern Ireland, Iraq, and Afghanistan.

- *Diffusion processes* are particularly suitable for modeling large-scale expansive conflicts across conflict regions, such as Mali, Iraq or Afghanistan (Zammit-Mangion et al., 2012). Driven by geographical attractors, conflict can tip in one or another direction, represented by a dynamic diffusion map with abrupt tipping behaviours.

- *Bi-stable models* already exist in animal ecology, where environmental factors such as food supply and temperature modulate whether insects fight or cooperate (Pruitt et al., 2018). Tipping between behaviour states is influenced by ecological changes and delayed recovery to prior states whereas increased variance is an early-warning sign of instability.

## 5. Modeling tipping cascades in conflict and cooperation

### 5.1 Bi-stable tipping models in action

Logistic and bi-stable models described in Sect. 4 are applicable here to tipping processes in human conflict and cooperation. An example are urban conflict models where quantitative and qualitative factors contribute to the tipping process when a group of people take a new trajectory towards violent mechanisms (Moser and Horn, 2011; Beall et al., 2010). This can be influenced by conventional factors (socio-demographic, ethnic/religious/caste, crime categories, legal framework); short-term tipping triggers (economic, political and media events); and long-term tipping bias (unemployment, parental guidance, substance use, etc.). For instance, urban violence is studied for several case studies such as the gender-based violence in Santiago and political violence in Nairobi (2008), factional violence in Dili (2006) and Sudan (2011), and Patna riots to improve security in the city. Such bi-stable models are used below as part of a larger networked model to reflect global connectivity (Aquino et al., 2019).

Bi-stable models are visualised here by landscapes of potential functions ("energy") in which objects are driven by internal and external forces. They are one way to conceptualize tipping between conflict and cooperation as two stable states, whereby

switching between them needs a certain amount of "extra" energy or incentive. We can regard those nations that take very little incentive to move from cooperation to conflict as fragile, and those that take a lot as being resilient. Let us briefly review alternative models of lower and greater complexity that can model state transitions (see Fig. 3):

1. *Discrete binary flip model* (e.g., Ising model): has independent variables that contribute to a probability of flipping between conflict and cooperation states, where the discrete model may not capture the tension and sliding dynamics.

2. *Continuous attractor model* (e.g., Potts, Kuramoto) with certain state(s) and independent variables that can push or pull the state between conflict and cooperation, e.g., the current state can transition and stay between attractor states.

3. *Continuous bi-stable model* (e.g., tipping point or logistic map) has two stable states and entropy wells (or basins of attraction), that entrap an agent within it.

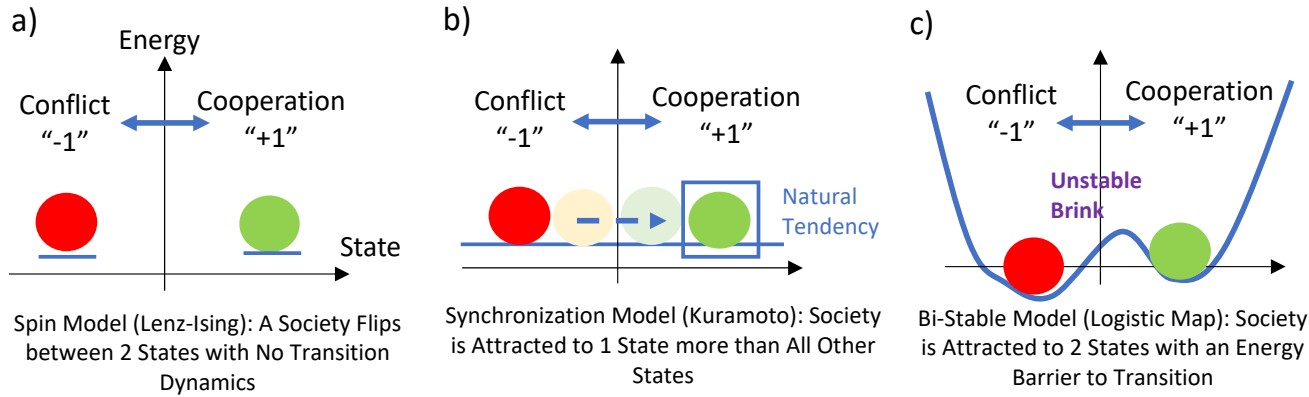

Figure 3: Concepts in modeling state transitions: (a) flip models, (b) attractor models, and (c) bi-stable models.

The third kind of tipping model is used to model the choice between cooperation or conflict under different environmental conditions. To create the simplest model that exhibits bi-stable tipping dynamics, we employ a third order polynomial for the

590 rate of change of state x, where the unstable brink is a tipping point:

$$\frac{dx}{dt} = \dot{x} = x\left(1 - \frac{x}{C}\right)\left(\frac{x}{K} - 1\right) + F \qquad (1)$$

Here, we are saying the rate of change of the dependent variable *x* (e.g., level of cooperation), is dependent on the current value of *x*, attracted towards the equilibrium state of cooperation *C* and a smaller conflict term *K*. An external forcing term *F* can include many factors such as trade, political influence, online influence, and climate change. In the following, we explore

some critical states in the model (as shown in Fig. 4):

- *Growth:* the greater the current level of cooperation *x*, the more cooperation occurs, subject to the limits of *C* and *K*.
- *Full Cooperation* (Stable State): when the level of cooperation reaches maximum capacity (*x*=*C*), the rate of growth is 0, meaning any further growth will naturally retract back to *C* value.
- *Minimum Cooperation* before Conflict (Unstable Tipping Point): when the level of cooperation reaches minimum

criticality (*x*=*K*), the rate of growth is 0, meaning any further decline will retract to conflict.
- *Conflict* (Stable State): when the level of cooperation reaches zero (*x*=0), the rate of growth is 0, meaning any further growth in cooperation will retract naturally back to conflict.

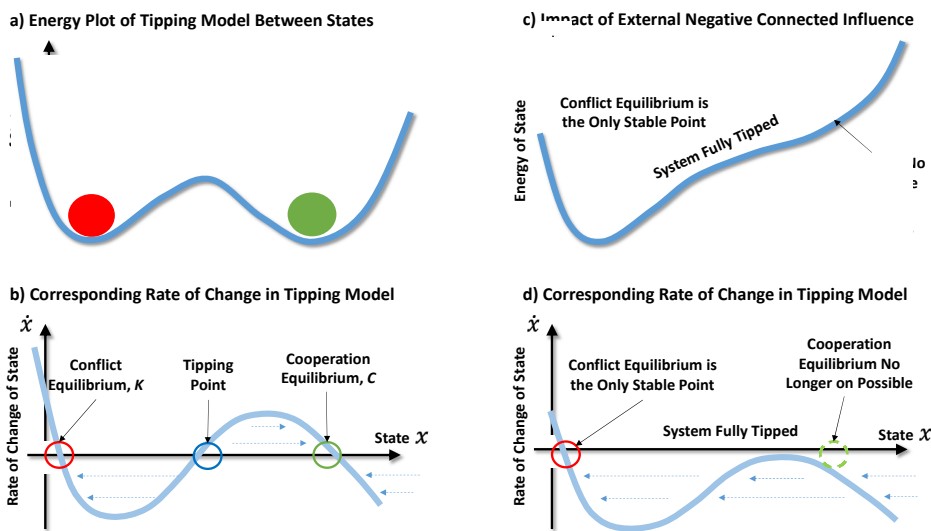

**Figure 4: Transition between stable states via tipping point: (a, c) energy plot of tipping model under different circumstances of no tipping and fully tipped, and (b, d) corresponding rate of state change diagram.**

The direction of growth towards cooperation (C) or decline towards conflict (K) is indicated by the light arrows pointing towards or away from these states. Fig 4a-b shows the corresponding energy and rate of state change plots for a standard tipping model. Fig. 4c-d shows the corresponding energy and rate of state change plots for a heavily tipped model to conflict.

## 5.2 Cascading and self-enforcing dynamics in conflict

### 5.2.1 Stable state transition scenarios

State transitions between conflict and cooperation can occur when sufficient energy drives the process. In Fig. 3c, we showed the equation of bi-stable systems, and the state transition between stable states via the tipping point (unstable brink in between). In Fig. 4c-d, we see that when a system is fully tipped due to internal behaviour, external forces or other reasons, stable cooperation states can disappear and the only stable point is conflict. Indeed, whether systems decide to be cooperative or in conflict can depend on many factors, which gives rise to "ambiguity" in literature where even fragile states can decide to cooperate or not depending on external support. As such, we are motivated to build a network of bi-stable systems to use the graph links to represent relationships between social systems (e.g., cities or countries).

### 5.2.2 Networked tipping points case study: the GUARD Project

One way to capture diverse external forces between different nodes is to construct a multi-layer graph (Fig. 5) with:

- *Nodes*: Cities or countries.
- *Links*: Relationships are links between nodes which can have either binary data (1 or 0), or weighted data (strength of relationship). The data can be dynamic to reflect change over time and be directional to reflect unilateral relations.
- *Graph Layers*: Multiple layers can reflect different types of relationships (diplomatic, military, transport, trade), and each layer can be interconnected to represent the strength of mutual coupling or cohesion.

**a) Multi-Layer Cascade Tipping Dynamics Network Model**

Nation Plane Network
(e.g., Political
Relationships)

Cultural Plane Network
(e.g., Religious Belief
Differences)

Geographic Plane
Network (e.g., Trade
Route Data)

Conflict
"-1"   Cooperation
"+1"

Each City Node is
Modeled with
Tipping Dynamics

**b) "GUARD" Model & Prediction Framework**

Networked Tipping Model Legend
Network Relations &
Climate Push and Pull

No Conflict
Data
High Conflict Data

Training
Data

Figure 5: (a) Multi-layered graphs that connect local nodes (city) tipping dynamics with inter-city relationships across geographic, cultural, and national/political relationship layers. (b) Example results taken from the UK Alan Turing Institute GUARD project (Aquino et al., 2019). Legend: each node (city) can tip between cooperation (triangle point upwards) or conflict (point downwards). The size of the triangle indicates network cascade vulnerability, and the colour indicates validation data (red is conflict, yellow is lack of conflict data). The link colours represent multi-layer data, where blue indicates some evidence of positive cooperation and green is the absence of evidence of trade and friendly relationships.

As a result, we create a multi-layered graph (see Fig. 5) with $N$ settlements. The connected tipping dynamics for node $i$ can be represented as:

$$\frac{dx_i(t)}{dt} = x_i \left(1 - \frac{x_i(t)}{c_i}\right)\left(\frac{x_i(t)}{K_i} - 1\right) + \sum_j^N A_{ij} g\big(x_i(t), x_j(t)\big) + F_i \qquad (2)$$

where the new summation term represents the graph connections (via connectivity matrix $A$) and the coupling data or function $g(.)$ between attributes in node $i$ and other attributes in nodes $j$. These graphs shown in Fig. 6b can be very large, in the Global Urban Analytics for Resilient Defence (GUARD) project (Aquino et al., 2019) we have $N$=7,000-50,000 settlements, and 200,000 to 1,000,000 consequential relationship links. We use historical data to learn the parameters of the model above by fitting independent variables to the dependent variable $x$. Here, conflict data $x(t)$ at time $t$ per node are used to fit with independent variables: previous historical state of $x(t-1)$ and the weight of graph connections to the node as independent parameters. Equation (2) describes the nonlinear relationship for change of state $x$, as well as the graph connections with other nodes via the multi-layer land transport connection matrix $A$, with friendly ties based on existence of economic or political treaties or military exchange/trade (1 or 0), and cultural similarity based on a religious belief vector of major religions (distance between vectors). The independent parameters are weighted by the $g(.)$ function determined by multi-variate regression. The data are from 2001 to 2017, and the conflict data (x) are from the Global Terrorism Database (GTD), where trade and transport data are from different UN, CIA, and National Geographic databases.

### 5.2.3 Cascading and self-enforcing dynamics in cooperation

We offer a brief review of techniques used to analyze a networked dynamical system, without going into too many details.

o    *Synchronization:* This is defined by whether the whole network of bi-stable nodes can reach a single (world peace), or multiple undefined states (e.g., different alignments). Part of the proof lies in whether this even exists as it is not guaranteed for every part of the network, and a perpetual flux of state transitions is very likely, especially under dynamic perturbations.

o    *Stability:* This is often defined by stable states that can be reached, how rapidly can they be reached, given some initial conditions, and how stable they are to further perturbations.

o    *Uncertainty:* This is of interest when we wish to understand how robust the model is to uncertainties in data, modeling approach, and further perturbations in the forces. A typical approach is to perturb the links or external forces such that the cascading impact can be examined (see Fig. 6a).

o    *Stochastic Resonance:* This may be of interest, when micro-oscillations can cause a state transition (Gammaitoni et al., 1989), for instance connected to small wet-drought patterns (see Section 6.1), or sub-threshold tension.

The technical challenge is the high-dimensional nature of the problem (e.g., the number of tipping equation parameters exacerbated by the size of the connecting graph), and the different dynamic perturbation combinations (see Fig. 6a). At the
simplest level, we assume that all nodes are at or near one of the stable states. This allows us to apply *mean field* approaches (see Fig. 6b, case i), of which the state of art is a sequential heterogeneous mean field (Moutsinas and Guo, 2020) to estimate the likely alignment between different states across the network. When we do not assume what stable state or transition between states each node is in, we must properly analyze the full dynamic possibility of the system, and this leads us into the region of attraction (RoA) method (see Fig. 6b, case ii) (Moutsinas et al., 2021). In Fig. 6c, we show how increased negative
perturbations on links can pull the system towards less cooperation states and more conflict, eventually a collapse to large-scale conflict difficult to bounce back to cooperation, which we often call a loss of resilience (Moutsinas & Guo, 2020).

**a) Network Perturbation Scenarios: (i) Link Removal, (ii) Negative Influence, (iii) External Forcing**

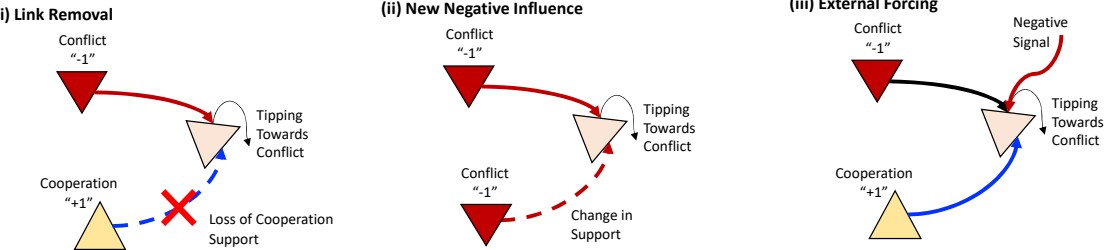

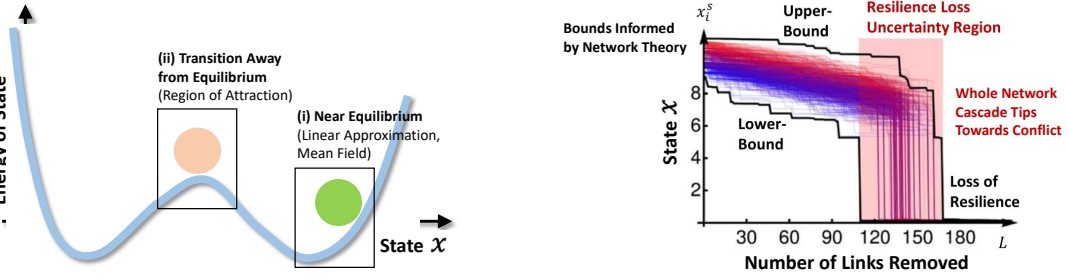

**Figure 6: Showing how network nodes can change their states (cooperation or conflict) based on external disturbances. Dynamic state transition analysis methods consist of: (a) network perturbation analysis, and (b) cascade analysis**
**method based on how far the state moves (Moutsinas et al. 2021), (c) impact of link removal on cascading transition of cooperation across network.**

## 6. Regional case studies for tipping cascades in conflict and cooperation

Following the climate-security discussion in Sect. 3, the review of tipping models in conflict and cooperation in Sect. 4 and the analysis of the transition's dynamics in the bi-stable tipping model of Sect. 5, we will now have a discussion of exemplary cases in regional hot spots of climate security. While this refers to key conceptions and model frameworks introduced earlier, we do not aim for a full model application of regional examples, but for illustrative cases of future in-depth research.

### 6.1 Transitions between conflict and cooperation in regional hot spots

The "risk multiplier" role of climate change is particularly relevant in vulnerable hot spot regions, such as the Mediterranean and Arctic region, the Sahel and Middle East, South Asia and Central Asia. Here it combines with other problems, including local degradation of ecosystems and land, absence of early warning and disaster protection, poverty, and political instability. Most vulnerable to climate stress are regions whose economies depend on climate-sensitive resources (water, food, forests, farmlands, and fishery) and where infrastructures are exposed to climate change, with a high dependence on agriculture, coastal areas and river basins. For the most severe consequences, adequate assistance is hardly possible and social systems become overloaded in the regions of concern. If people cannot cope with the consequences and limit the risks, tipping cascades to instability and conflict may be more likely and propagate through systemic networks like a domino chain. External aid and internal cooperation between affected communities can influence the regional tipping mechanisms in one or the other way. We use selective examples from three different regions to identify key mechanism inducing or preventing tipping in conflict.

The Syrian civil war has been considered as a striking case for social tipping points and related cascades of security risks and conflict. Before the Syrian uprising in 2011, the most severe drought was recorded in the Fertile Crescent which according to Kelley et al (2015) contributed to the loss of livelihood of farmers in Syria, a link that remains contested (Selby et al., 2017), as well as the role of agricultural losses or the rural to urban migration (Ide, 2018). Compounding with other conflict drivers (Arabic Spring, neoliberal reforms, dissatisfaction with Assad regime, aftermath of Iraq war and Islamic State, bad governance, military response, forced displacement) the situation erupted in a highly violent conflict. The escalation emerging from the compounding conflict dynamics spread to neighbour regions and beyond, involved rivalling powers and moved a large number or people towards Europe. This resembled a tipping cascade of interacting multiple drivers, challenging European politics in a polycrisis of displacement, terrorism, nationalism, populism and other crisis drivers beyond Syria.

While South Asia is facing low development, dense population and a substantial dependence on agriculture exposed to climate change (Wischnath and Buhaug, 2014), the number of armed conflicts has been increasing from 1850 in 2000 to 2846 in 2015, where most events occurred in Afghanistan, Sri Lanka, Nepal, Pakistan-Afghanistan border, and the seven Northeastern states of India (Xie et al., 2022). Quantifying the effects of climate variability on armed conflict in South Asia, Xie et al., (2022) found that precipitation can impact armed conflict via direct and indirect pathways which are contradictory in sign, while temperature affects armed conflict negatively through a direct path and indirect effects were insignificant. Involving the intermediary variables water resources, crop yield and income, net combined impacts are weak due to two contradictory effects offsetting each other which indicates no clear sign of self-enforcing tipping points. However under unfavorable weather conditions with substantial crop yield reduction and economic losses, violence might become a source of income for those who cannot live on rain-fed agriculture, food supply, or livelihood security. The complex contradictory interactions may also occur at country level where adaptation and mobility in agriculture can mitigate climate and conflict risk (Abid et al., 2025; Mobeen et al., 2023). When increasing risk of losing livelihoods due to sea level rise, salinity, droughts, storms and other hazards exceeds adaptive capacities, negative transformations may unfold which are not well understood. A case study

explored how smallholder villagers in Tamil Nadu, India, are managing systemic livelihood risk (Mechler et al., 2019) by sea level rise, coastal inundation, droughts (as slow-onset events) and flooding by cyclonic storms. Risk analysis and impact chain assessment together with surveys and participatory engagement with households and institutions found risk management actions to secure their livelihoods (rice cultivation, sea dikes, freshwater tanks, salt-tolerant paddy seeds, non-farming labor), preventing tipping points and inducing deliberate transformational change (Mechler et al., 2019; Juhola, et al. 2022).

## 6.2 Case study: the Lake Chad region

The Lake Chad region has experienced some of the most striking social and bio geophysical changes in recent times. Just 50 years ago, the lake was larger than the size of Israel (25,000km$^2$) and provided livelihoods to over 30 million people (Gao et al., 2011; Okpara et al., 2015). Over recent decades, Lake Chad has been facing strong variability in lake water levels and decline in water flows from rivers, rapidly rising temperatures (1.5 times faster than global average), longer dry seasons, heat waves, and sand/dust storms, have contributed to crop failures, livestock losses, and depletion of fisheries, and placed the region on the edge of systemic criticality and conflict tipping pathways. At the same time, the region has been afflicted by several political, identity, ethnic, communal, and resource conflict events, some of which have tipped over into massive upheavals in the form of terrorism, triggering brutal violence. Conflict tipping into violence under conditions of rapid lake water oscillation and shrinkage has triggered a shift from a state of relative tension to a heightened violent situation where self-perpetuating cycles of open violence become more prevalent and harmful to the Lake Chad biogeographical and ecological landscape (Avis, 2020). Conflict tipping pathways in this setting are diverse and multifaceted.

One conflict tipping pathway is the abrupt breakdown in small-scale farming, fisheries, and local food systems triggered by multi-year oscillations of the Lake Chad waters (Okpara et al., 2017). This has resulted in massive wellbeing deficits and amplified social grievances against the state. Grievances have fuelled the formation of violent solidarity networks (many with links to criminal gangs and insurgent groups) and have led to brutal regional conflicts and the death and displacement of millions of citizens. Another tipping pathway is the escalation of a conflict economy where armed groups illegally control natural resources, agricultural trade routes, and food supply chains, and secretly divert arms, drugs, stolen cash, and cattle into areas they control (Sampaio, 2022). Armed groups recruit and radicalize young fighters who previously depended on the resources from the Lake. In doing so, they trigger spiralling territorial dynamics where the intensity and scope of conflict and violence rapidly increase. At the same time, cycles of retaliation, reprisals, and counterattacks between state and non-state actors (linked to the conflict economy) have continued to create self-perpetuating chains of violence.

Conflict tipping over into violence and terrorism harm the Lake Chad biogeographical landscape in many ways. For example, approximately 80% of the conflicts take place in nature-rich, biodiversity hotspots, and with the increasing use of the environment as a hideout, military base or camp for hostage taking, attacking the environment has apparently become a military/warfare objective (Okpara et al., 2015). Aerial and ground bombardments by soldiers primarily target the inland hardwood forests and the mangroves covering remote insurgent groups' camps, causing direct environmental damage. Similarly, the intentional bombing of villages, markets, religious centres, schools, power plants, and telecommunication facilities by insurgent groups produces many hundreds of thousand tons of emissions in carbon monoxide, nitrogen oxides, hydrocarbons, Sulphur monoxide, and $CO_2$, which adversely impact human, plant, animal, and bird populations in the region. Bomb particles contaminate water supplies in communities, undermining public health.

Conflict tipping also induces indirect harm to the Earth system. Conflict tipping triggered population displacement and complex emergencies in the region, led to overcrowding in destination areas, and intensified pressures on regional water, food,

land, and energy systems (Vivekananda et al., 2019; Oginni et al., 2020; Kamta et al., 2021). These outcomes in turn spurred unsustainable agricultural practices, overfishing, and deforestation. Displaced people have turned to the environment to meet their basic needs - woods are removed regularly from forests to build shelters, make fire for cooking and heating, and to create charcoal for sale. Displaced people take on hunting expeditions which threaten animal biodiversity; and the wastes they produce contaminate land and water resources. Lake Chad conflict tipping (under displacement crises and growing conflict economies) is characterized by a breakdown in environmental laws and governance, causing weak enforcement of nature conservation mechanisms (Magrin, 2016). Increased illegal logging, poaching, and resource exploitation resulting from this has further exacerbated environmental degradation.

An example of change in behaviour that is hard to reverse is the motivation amongst young people to embrace extremism (fuelled by abrupt breakdown in livelihood services). By generating income through participation in regional conflict economies, young people have now built capacity to defend rebel groups, seeking opportunities to perpetuate violence . This is made worse by the climate crisis and has become more widespread despite recent rebound in the Lake waters (Pham-Duc et al., 2020). Gradual variations of water level have added a new twist: communities that moved and built homes towards the dry and small Lake Chad during the droughts of 1980s and 1990s are now having to confront massive flooding (particularly during rainy seasons and any time the Lake overstretch its banks); many have lost their natural and physical assets (land, farms and houses) as the Lake expands, rebounds and recovers (the Lake is somehow reclaiming back the land areas it initially lost). We conclude that several mechanisms of the tipping point definition (self-perpetuation, substantial, widespread, often abrupt and irreversible impacts) can be found in Lake Chad region which combine in vicious circles between violent conflict and environmental degradation, including breakdown of livelihoods, oscillations of Lake Chad water, chains of violence and displacement, and rebel-controlled conflict economy, without one single cause.

## 6.3 Bi-stable tipping model in Lake Chad region

Based on the bi-stable model in Sect. 5 we demonstrate how in the Lake Chad case study meso-scale groups and communities can exhibit diverse and dynamic behaviours when under climate stressors. Fig. 7 is pictorial narrative on how tipping models can give different tipping dynamics based on how they reinforce or compete with each other through a network. In Fig. 7a, we see that a macroscale aggregate (e.g., nation, region), can comprise several meso-scale communities including individuals. Each community's behaviour (or risk of behaviour) can be represented by the tipping model in Sect. 4, where there are two stable states (conflict and cooperation/peace). In these tipping dynamic systems, they are primarily affected by two variables:

- Internal parameters affect the fragility of tipping, e.g., how easy it is to transition between states.
- External forcing affects the tilt, biasing the likelihood of tipping one way over the other.

Here, we can see several cases (Figure 7a):

- ➢ *Blue community:* poor governance can lead to a very low barrier, allowing easy transition between conflict and cooperation (vulnerable to external perturbation)
- ➢ *Yellow community:* climate change can tilt dynamics in favour of conflict, but a healthy societal resilience (reflected in the barrier) can potentially keep the community in peace.

In Fig. 7b we can see an example community in Lake Chad, whereby it first experiences drought which seems to drive the social system to tip towards conflict, but this is not enough. The first enabling pathway is migration of fishing communities away from Lake Chad, which decreases the barrier and increases the opportunity for conflict. This opens the second enabling pathway which is that the power vacuum allows militants to move in and create conflict. The conflict between militants and remaining communities in turn raises the barrier for any return to peace, so even a return to wet season means fishing communities cannot easily return and restore peace. This relates to the narrative of Lake Chad in Sect. 6.2.

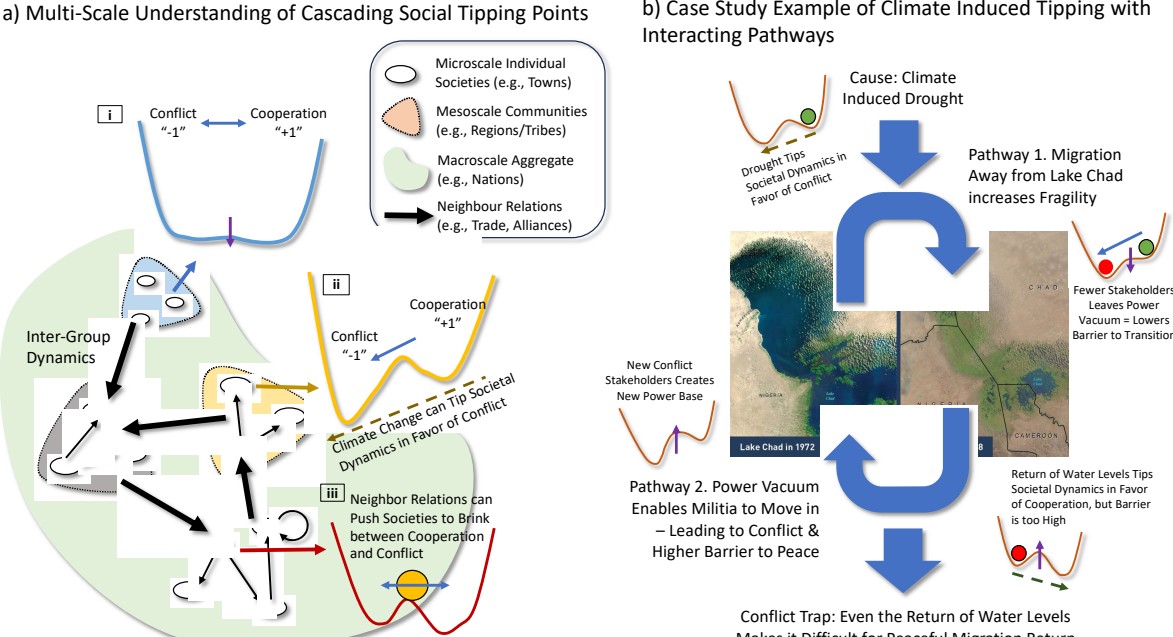

**Figure 7: (a) Multi-scale understanding of cascading transitions between conflict and cooperation/peace, and (b) Example case study in Lake Chad region of climate induced tipping points with interacting pathways.**

## 7. Governance challenges

### 7.1 Managing negative and positive tipping

Climate-induced conflicts require adaptive and anticipative governance (AAG) to effectively prevent and contain negative tipping to violent conflict and induce positive tipping towards cooperative solutions and synergies of climate-security risks. To stabilize climate-society interaction under deep uncertainty and complexity, diverse knowledge and deeper understanding is essential for researchers and decisionmakers about the underlying processes, how they interact and can be influenced. Besides data and experience theories and models can also contribute to tipping management and governance, including identification of drivers and barriers of tipping points, their temporal and spatial windows and conditions for stability and reachability, detectability and controllability. Appropriate indicators and instruments support adaptive decisions and responses to past, current and future tipping cascades in Integrated Earth system models.

Model scenarios can indicate tipping to conflict and how to prevent and contain escalation by political and legal measures, strengthen resilience and cohesion of societies in the long run, by enabling communities to respond to combined climate and conflict risk and avoid polarization. Managing tipping cascades considers the respective scales of decision-making, from micro to macro, which differ from the traditional sector-specific approaches of vulnerabilities and feedback mechanism. Moving closer to windows of potential tipping, more reliable information is needed to prepare, prevent and adapt. AAG benefits from monitoring and early warning systems (EWS) that detect and indicate signals of disaster and conflict before they occur, finding pathways for desirable change (e.g., Haasnoot et al., 2019). Conditions for successful EWS are (Grimm and Schneider, 2011): (a) systematic collection of event data and expert assessments; (b) data analysis based on advanced social science techniques; (c) strategic response scenarios and consequence assessments; (d) presentation and implementation of options to policymakers. EWS can facilitate preventive responses across multiple levels, networks and stakeholders to establish regulatory regimes and institutions when needed (Juhola et al., 2022), including management of disaster, crisis and conflict to limit damages.

To understand destabilizing tipping cascades, systemic risks, hazardous pathways and acceptable boundaries, different techniques can be integrated, involving system models, ABM, SNA and AI/ML to guide AAG actions. A stronger stakeholder engagement might help to design participative modeling interfaces, considering social equity and justice, as well as dimensions of political economy, power and distribution. An example is the now commercial platform Global Urban Analytics for Resilient Defence (GUARD) using data, such as the GTD/UCDP/PRIO Armed Conflict Dataset together with advanced AI/ML systems (Guo et al., 2018; Ge et al., 2022; Xie et al., 2022). Results can feed into institutional frameworks, such as the UN Climate Security Mechanism.

Whether climate stress drives a system from undesirable to favourable pathways, from conflict to cooperation, or from vicious or virtuous circles, pathways and opportunities for action consider enabling and constraining conditions. Diverging and competing pathways illustrate how interacting choices and actions by diverse governments, private sector and civil society can advance climate resilient development, shift pathways towards sustainability, and enable lower emissions and adaptation. There is a narrowing window of opportunity to secure a livable and sustainable future for all (IPCC, 2023). The longer mitigation is delayed, the less effective are adaptation options.

Besides predicting and avoiding negative tipping, governance opportunities for positive (desirable) tipping cascades are essential, based on norms and goals to be achieved, such as the sustainable development goals or staying within the planetary boundaries, including the climate targets in the Paris agreement. To bring a system to an intended tipping point requires some "forcing" for transformation. Since agents select actions that are more beneficial, less costly and less risky compared to alternatives, managing positive tipping cascades could apply ABM approaches such as the VIABLE framework to represent motivations and capabilities in switching to alternative pathways (in energy, food, health, etc.) and overcome path dependency, lock-ins, time discounting and established habits. For instance, in the energy transition the strong dependence on fossils requires policies to increase the benefits of renewables (e.g., by subsidies or sharing the revenues) and reduce their cost, risk and conflict potential such that a critical mass is built for a self-enforcing collective positive cascade. Human agency can intentionally utilize trigger-response mechanisms and feedbacks in social systems to establish new norms and collective action, across scales and social systems. New adaptation practices go beyond incremental adjustments toward transformational adaptation involving systemic change (Juhola et al., 2022). Diverse sources of knowledge can help to contain this uncertainty, including scientific data and modeling as well as local and indigenous knowledge based on experience, mobilized in participatory approaches and collective learning. Agency benefits from constructive and mutually adaptive behaviour of agents to induce positive tipping in swarming interactions.

**7.2 Conflict transformation and environmental peacebuilding**

Civil conflict transformation (CCT) is nonviolent and uses only civil measures. Mediation, victim-offender mediation, conciliation, round tables and dialogue forums are prominent examples, as well as other forms of violence prevention and diplomacy, peacebuilding and civil peacekeeping. CCT can make an important contribution to dealing with climate change and the current crises of nature-society relations. Climate policy can be used as peace policy and vice versa (Pastoors et al., 2022). Conflict tipping has the potential to foster creativity and community and can create positive transformative opportunities to reconstruct societies. Conflict transformation and cooperation can facilitate this process, especially when a critical mass of conflict actors adopts new attitudes, behaviours or values. This can lead to positive social tipping effects that spill over into society. Conflict transformation deals with and transforms the immediate and deep issues people fight about, such as marginalization, abuse of human rights and widening acute poverty. By creating inclusive and equitable social structures or

fostering paradigmatic events that shift the perceptions and attitudes of conflict actors toward dialogue, trust building, and reconciliation, conflict transformation can move conflicting parties from a state of confrontation and violence to one of cooperation which combined with conflict transformation can create powerful synergies with positive changes in society.

One approach to foster such conflict transformation towards cooperation is environmental peacebuilding, which focuses on managing renewable natural resources in conflict-affected areas with a conflict-sensitive and sustainable approach, aiming to promote lasting peace (Krampe et al., 2021). Research in this field is divided into two main schools: cooperation perspectives and resource risk perspectives. Cooperation perspectives prioritize collaborative approaches to managing environmental resources, aiming to promote peace-making through spill-over effects and positive peace outcomes that encompass human security and equity. In contrast, resource risk perspectives emphasize resource-induced instability and sustaining negative peace through environmental cooperation (Krampe, 2017).

While quantitative studies have been conducted, there has been a notable increase in qualitative case studies, particularly emphasizing the local and everyday experiences of environmental peacebuilding (Ide et al., 2021; Johnson et al., 2021). This localized perspective is crucial, as key dynamics of environmental peacebuilding often occur at the local level. Recent studies highlighted the need to advance scholarship to better theorize the causal understanding of natural resource management in post-conflict settings (Johnson et al., 2021, Krampe et al., 2021) and of providing proof of the contribution of environmental peacebuilding initiatives to positive peace and mitigate tipping points. Krampe et al., 2021 have proposed three theoretical mechanisms to explain how environmental cooperation can contribute to positive peace: (1) the contact hypothesis posits that facilitating intergroup cooperation reduces bias and prejudice; (2) the diffusion of transnational norms suggests that introducing environmental norms supports human empowerment and strengthens civil society; and (3) equitable state service provision, which argues that providing and ensuring access to public services that address the instrumental needs of communities strengthens state legitimacy and state-society relations.

As interventions become more common, there is a growing recognition of the negative implications of environmental peacebuilding, often referred to as "backdraft", which are linked to maladaptation (SIPRI, 2022). Emerging research sheds light on the unintended and unanticipated consequences of environmental peacebuilding interventions (Ide, 2020). Further research in this area is necessary to avoid romanticizing the concept of environmental peacebuilding and to learn from past experiences to prevent potential conflicts triggered by poorly planned and managed climate adaptation and mitigation efforts. Ideally, this research can also help identify opportunities to generate positive legacies of peace (see e.g. Simangan et al., 2023).

## 8 Summary and conclusions

This study connects the wealth of empirical research on the dynamics of conflict and cooperation under climate change with the growing research on tipping points, compounding and cascading risks, in the qualitative discussion of complex pathways, transitions and interactions of tipping in security-related issues and by a selective review and application of quantitative modeling, connecting both research fields in a novel way. Following an introductory contexualization in complexity science and today's multiple crisis landscapes, a structured overview (Sect. 3) highlights that climate and environmental change can affect the relationship between conflict and cooperation, depending on certain conflict-relevant conditions and multiple risk indicators which affect human choice and societal responses triggering tipping via key pathways in the climate-conflict nexus (livelihood deterioration, migration and mobility, opportunities for militant and armed actors, grievances). Compounding effects can result from the double exposure to conflict risk and environmental vulnerability, making it difficult to separate them in a mutually enforcing and escalating vicious circle beyond critical thresholds.

Within the framing of integrative Earth system dynamics, relevant model types of conflict and cooperation are introduced in Sect. 4. Among more specific models we focus on system models analyzing dynamic trajectories, equilibria, stability, chaos and empirical applications to conflicts (such as the Richardson model), as well as agent models following decision rules of behaviour based on motivation and capability, driving and preventing conflictive or cooperative actions. The VIABLE model framework integrates the adaptive dynamics of values, capabilities and priorities for systemic actions, providing simulations of conflict cases and analytic conditions of social equilibria, stability and complexity of multi-agent interaction and related tipping, cascading, networking and transformation.

An illustrative bi-stable tipping model in action is presented and applied in Sect. 5, to study the cascading and self-enforcing dynamics involving capacity, criticality and external forcing, and present transition scenarios between states of conflict and cooperation which depend on the levels of stabilizing and destabilizing forces. The tipping dynamics is shaped by internal factors, such as fragility and resilience, and by external factors affecting the tilt, such as trade, climate change, political and media influence. The effect of city or country networks is represented by multi-layered graphs connecting local and network tipping dynamics which show how negative perturbations on links can reduce resilience and pull the system towards less cooperation or collapse to large-scale conflict.

Within the modeling context a regional hot spot perspective on the cascading "risk multiplier" role of climate change is pesented in Sect. 6, using Lake Chad as a case study. Based on the bi-stable model we demonstrate how in a macroscale aggregate of nations and regions, meso- and small-scale groups and communities can exhibit diverse and dynamic behaviours under climate stress between conflict and cooperation/peace. Discussed is the effect of internal parameters on fragility and the likelihood of tipping, and how external forcing affects the tilt. For poor governance community behaviour is facing a low barrier against transition, such that vulnerability to climate change can tilt dynamics towards conflict, but a healthy societal resilience serves as a barrier keeping the community in peace. Narratives that droughts can tip the social system towards conflict, need to be contextualized with other pathways decreasing the barrier such as migration of communities away from Lake Chad and a power vacuum allowing militants to create violent conflict which raises the barrier for return to peace. Thus conflict tipping to violence and terrorism further undermines the chance for cooperation and exacerbates environmental degradation. Finally, Sect. 7 discusses how governance can prevent and contain climate-induced tipping to violent conflict and induce positive tipping towards cooperative solutions of security risks. Adaptive and anticipative governance contribute to institutional mechanisms for cooperative security, civil conflict transformation and environmental peacebuilding, creating synergies for sustainable peace. A better scientific understanding of the complex interactions contributes to forward-looking cooperative policies to prevent violent conflict and enable stabilization of the Earth system.

*Competing interests:* The contact author has declared that none of the authors has any competing interests.

*Acknowledgement:* Authors acknowledge support by the Cluster of Excellence "Climate, Climatic Change and Society" (CLICCS) funded by Deutsche Forschungsgemeinschaft (J. Scheffran), and the USAFOFSR Networked Social Influence and Acceptance in a New Age of Crises (FA8655-20-1-7031), and Alan Turing Institute (D&S) GUARD Project (W. Guo), including a UKRI Future Leaders Fund (Grant No: MR/V022318/1) offered to Uche Okpara.

*Author share of work:* J.S. coordinated and edited the whole manuscript, all authors designed, discussed and edited it. Most significant contributions to individual parts were made by J.S. (Sect. 1, 3, 6), W.G. (Sect. 4, 5), F.K. (Sect. 2, 6), U.O. (2, 5).

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
