# Peer review of "Tipping cascades between conflict and cooperation in climate change"

_EGUsphere, 2023_

## Author Comment (AC1)

**Tipping cascades between conflict and cooperation in climate change**

*Jürgen Scheffran, Weisi Guo, Florian Krampe, Uche Okpara*

**Reply to Review Comment (RC1)**

**Summary:** This is an interesting but strange paper. Part of the text is very technical and hard to follow for the average social scientist, and the link to real-world challenges related to conflict and cooperation under climate change is sometimes lost. I appreciate the attempt to bridge perspectives across scales and drawing on a range of methodologies, but due to incomplete integration of these perspectives, I believe the present version of the paper does not fully deliver on its objective "to improve the understanding of the current issues of conflict and cooperation, in particular in climate and environmental and conflict…" (p. 2). It also is too long.

**Response:** Thank you for the constructive comments which are greatly appreciated. To our knowledge this is one of the first papers that brings research on climate and conflict together with research on tipping cascades in conflict and cooperation. While the first field has been addressed mostly by quantitative statistical methods of large-scale data or case-based qualitative research, the second field is rooted in conceptual and modelling approaches of tipping points and cascading events, including system and agent-based models. Aiming for an interdisciplinary approach connecting the communities of social and natural scientists, bridging both perspectives across scales and methodologies is challenging, as the reviewer notes, but the attempt is also promising in a world where multiple crises aggregate into a polycrisis and one discipline or method alone cannot address the complexity of interconnected issues. We will take more effort to bring the different perspectives together and demonstrate the relevance of tipping between conflict and cooperation.
We admit that more could be done to overcome the limits of "incomplete integration". One way is to further develop and expand the framework of pathways of climate-security interaction (Figure 1) to better explain and integrate the different parts and pathways as well as tipping cascades and connect to real-world problems. We provide some ideas and leave more space for inspiring the community to imagine the opportunities in their own way. To follow the comments we aim for a better balance of the different sections and the integrative connections between them, in particular extending the climate-conflict review (Section 2) with providing exemplary cases including cooperation, a shorter review on models of tipping in conflict and cooperation (Section 3) and giving more explanations and details in the result Section 4 and case study Section 5. We are also open to shortening Sections 3 and 4. For example, we can move some of the technical details to an Appendix in response to you concerns, to facilitate the flow of the main text and preserve details for readers who look for approach and evidence.

**Comment 1:** The introductory section 1.2 on concepts was very useful. However, it can be improved further by drawing on concrete examples of tipping elements and cascades in the social sphere as well; presently examples are given predominantly from climate / natural systems. What are good real-world examples of exponential chain reactions through social systems? I also think a brief reflection on the role of climate (incl. weather) versus climate change would be in order, and a cleared and more explicit separation between these distinct phenomena is wanted. To what extent are we able to distinguish between these sources of risk empirically – and what are implications for our ability to offer insights and advice on the nature, probability, and severity of future risks?

**Response:** Thank you for the encouraging note on linking concepts to real-world examples of tipping elements and cascades in the social sphere. To motivate the work earlier in the paper we would like to critically discuss the tipping potential and cascading mechanisms of our main case of Lake Chad (which is studied in more depth in Section 6) and other cases that may be relevant to society and conflict, such as the fall of the Berlin Wall and Arabic Spring, the Syrian civil war and Russia-Ukraine war, pandemic and climate effects on South Asia (for examples

see Lenton et al. 2023). It is also worth discussing and comparing connections between natural and social tipping points (Pruitt et al. 2018; Jin and Guo 2022).

As suggested, we will briefly distinguish between weather, climate and climate change as sources of risk and the implications for addressing future risks and conflicts, at temporal and spatial scales (see Dahm et al. 2023). Where weather is often short-term events (e.g. three consecutive hot days in summer) or an extreme event (e.g. storm or flood of certain intensity), we refer to climate change and systematic change of ambient conditions with long-term forcing effects and thresholds, which are often reflected in the mean or variance of data changing. Examples are the risk of a certain crop dying or a particular road washed away, or behavioural changes when outdoor labour becomes inefficient or conflict emerges more likely beyond a certain average temperature, as reflected in anomalies in a data stream rather than a constant change. In the tipping dynamics we consider climate change as a force on the social system, whereas weather is a short-term shock. Within the limits of this paper, we will discuss the interaction of forces across spatial and temporal scales while a comprehensive theory is beyond the scope of discussion.

**Comment 2:** In several places, displacement and mobility are mentioned as potential pathways to conflict and instability. Here, some consideration of counterfactual outcomes (e.g., staying put) would seem relevant. Mobility is an important risk management strategy – and especially so in dryland regions with large seasonal and annual variations in environmental conditions. While moving may entail exposure to new risks, it also typically reduces some sources of risk, so a holistic approach would need to consider both.

**Response:** Thank you for this important point which needs an extended perspective on mobility and migration. In the paper, we already highlight climate-conflict pathways through migration, mobility and other context-specific factors, with an example of the Lake Chad. While mobility is sometimes portrayed as a path to social harm under climate change and conflict, sparking divergent viewpoints and controversies, we agree that mobility can influence multiple outcomes. Alternative perspectives are important to integrate migration into adaptation and risk management strategies, as suggested by the reviewer and some of us on other occasions (e.g. Scheffran et al. 2012; Gioli et al. 2016), as well as in a recent PNAS Special Feature on "Migration and Sustainability" (Adger et al. 2024). We can refer to some of this work from a more holistic approach.

**Comment 3:** I found the language on pp. 5-6 on risk dynamics too deterministic and insensitive to context. At the same time, all given examples are from Sub-Saharan African countries / regions so presumably these pathways are not equally likely to play out everywhere, at any given time. (I'm sure the authors would agree to that.) It also reminded me of Adams et al.'s critique of sampling bias in this field, which is cited elsewhere in the paper I think but not reflected on here. Is there a general problem that we lack case studies of communities / locations / periods where a climate shock (tipping point) is observed but no resulting change in human behavior (tipping point) is documented? Lastly under this point, the discussion of pathways focuses almost exclusively on conflict, while cooperation also is supposed to be covered by the study. So perhaps add some examples of cooperative responses to various risk dynamics as well?

**Response:** Following your comment, to avoid making the "risk dynamics too deterministic and insensitive to context" we agree that risk pathways depend on regional conditions. We can expand on our statement on page 5 "The relationship between cooperation, climate change and security risks therefore varies depending on context, is often indirect and not linear." In particular, we emphasize here more contextual conditions and the role of agency which was already considered in Sections 2.4 and in Section 6.1. Regarding sampling bias we also refer here to Adams et al (2018) and consider cases beyond Sub-Saharan Africa and more examples of cooperative responses to risk dynamics, which we have already addressed in response to Comment 2. We consider potential connections between climate shocks and

economic losses with or without resulting tipping in human behaviour. This can clarify the distinction between tipping points, critical thresholds and economic shocks associated with tipping elements, gradual climate change, or non-climatic triggers (Kopp et al. 2016).
Our argument, which is emphasized across the paper, but perhaps not clear enough, is not that there is universality but dependency on specific circumstances. Just because we couple climate shocks with social tipping, it does not mean a certain shock will universally tip over society, because the parameters of resilience, cohesion and mutual support between societal communities are vastly different. Of course, there are cases where nothing happens (in terms of conflict), and in Figure 7a we show that certain societies may have a low barrier to transition, whereas others have high barriers from societal organisation or mutual support.

**Comment 4:** There is very little consideration of risk resulting from human / social responses to climate change. This is an opportunity missed. While doomsday scenarios of uncontrollable impacts of draconian geoengineering may be speculative and fanciful, what about protests to climate policies (think yellow vests) as well as protests to insufficient climate policies (Greta Thunberg…). With increasing warming, both these dynamics might be expected to become more prevalent and perhaps increasingly violent.

**Response:** This comment rightly points out that some of the many interactions between climate conflict and cooperation are considered in the paper but there are more. We can revise the manuscript and refer to the mentioned and other connecting pathways, including joint work by one of the coauthors (J.S.) on cooperation in conflict (Bukari et al. 2018), climate impact of violent conflict (Vogler et al. 2023), or conflict and protest over climate policies (mitigation, adaptation, disaster management, climate engineering) (Scheffran and Cannaday (2013) which need to consider conflict sensitivity (Nadiruzzaman et al. 2022).

**Comment 5:** Section 3 of the paper is too long, too detailed on historical evolution of models, and does not stick to the thread of the paper throughout. It would be good if the presentation of these models could engage more explicitly and concretely with the topic at hand: tipping points in climate-driven conflict and cooperation.

**Response:** We agree to condense the review on models of tipping in conflict and cooperation (Section 3) and focus more specifically on the essential aspects suggested (possibly in the form of a table), to classify the model types and their relevance to tipping cascades in conflict and cooperation. In response, we can also move some of the details into an Appendix.

**Comment 6:** Likewise, Section 4 remains too detached from the substance. It would help if the model could use specific (if arbitrary) values for the parameters to estimate conflict / cooperation outcomes under various assumptions, as opposed to leaving it as a hypothetical mathematical equation and illustrations that contain no information about climate stressors, social actors, specific risk metric (probability or magnitude of some outcome), etc.

**Response:** The core of Section 4 is to introduce and demonstrate the concept of bi-stability of two states such as conflict and cooperation between which tipping occurs beyond thresholds. This is based in complex systems science and an essential contribution of our paper to connecting conflict and cooperation with tipping point research. While not everyone may be familiar with related methods, the aim is to translate qualitative features of reality into the modelling world and its equations to build bridges across the communities. We use mathematical equation (1) as a basic representation of a bi-stable dynamic system that contains its main qualitative features and can be visualized in the graphs to show the intuitive meaning. Adapting the general model to real applications in climate and conflict-cooperation tipping cases is subject to future research based on case-specific. We can either cut this part short to reduce the complexity to key qualitative messages and minimum figures or provide more in-depth explanations and analysis of the underlying model and the data used.

*[We explained our approach in the response to Reviewer 2 and we copy that response here.]* To indicate the possible direction, we refer to the work by one of the coauthors (W.G.), building on a nonlinear dynamic model of conflict via interaction networks (see Aquino et al. 2019 and other sources, cited in Section 4). Here, conflict data $x(t)$ per city/town are used as the node level dependent variables at time $t$, to fit with independent variables that are: historical state of $x(t-1)$ and the weight of graph connections to the node as independent parameters. Equation (2) describes the nonlinear relationship between $x(t)$ and $x(t-1)$, as well as the graph connections with other nodes via the connection matrix $A$. The independent parameters are weighted by the g(.) function: (i) land transport connection ($A$ matrix: 1 or 0), (ii) friendly ties based on existence of economic or political treaties (1 or 0), and (iii) cultural similarity based on religious belief vector of major religions (distance between vectors). We use a multi-variate regression to find the weight of the independent parameters. The data ranges are from 2001 to 2017, and the conflict data (x) is from the Global Terrorism Database (GTD), whereas the trade and transport data is from different UN, CIA and National Geographic databases.

**Comment 7:** The discussion of Lake Chad is easier to follow, but I am not sure it helps us understand the relevance of tipping elements very well. Any observed change in behavior that is hard to reverse – e.g., the outbreak of violent clashes – constitutes a tipping point of some kind. But what is the challenge here is to assess the contribution of climate (variability or change) stressors (tipping point?) to that outcome, and I did not see much specific evidence of that. The presented causal narrative sounds reasonable, but how could it be evaluated? What would it take to conclude that the storyline doesn't reflect realities? One potential challenge to the narrative is the fact that the lake has not continued to shrink but rather show evidence of gradual increase in volume in recent decades, when violence has become more widespread. At least, the presentation of the hydrology of the lake should take account of this, although one might argue that it is inconsistent with assumptions about increasing climate-induced livelihood challenges. https://doi.org/10.1038/s41598-020-62417-w.

**Response:** Thank you for raising these questions about the justification of tipping points in the Lake Chad case which we can better explain, including Figure 7. According to a definition in the 2023 Global Tipping Point Report, a tipping point occurs "when change in part of a system becomes self-perpetuating beyond some threshold, leading to substantial, widespread, often abrupt and irreversible, impacts." Which of these mechanisms are actually met can be subject to discussion. Most important is the self-perpetuation of change which is often but not necessarily abrupt or irreversible. In the revised paper we can further discuss a few tipping elements associated with conflict, violence and terrorism in Lake Chad: (i) tipping in terms of abrupt breakdown in small-scale farming, fisheries, and local food systems triggered by multi-year oscillations of the Lake Chad water (potentially climate-induced); (ii) self-perpetuating chains of violence and displacement triggered by a rebel-controlled conflict economy.
As an example of change in behaviour that is hard to reverse, abrupt breakdown in chains of livelihood services motivates young people to embrace extremism. Creating income from conflict economies, they build capacity to defend rebel groups and seek opportunities to perpetuate chains of violence (observed change in behaviour). This is made worse by the climate crisis and has become more widespread despite recent rebound in the Lake waters (Pham-Duc et al 2020). Gradual variations of water level have added a new twist: communities that moved and built homes towards the dry and small Lake Chad during the droughts of 1980s and 1990s are now having to confront massive flooding (during rainy seasons and times the Lake overstretched it banks); many have lost their natural and physical assets (land, farms and houses) as the Lake expands, rebounds and recovers (the Lake is somehow reclaiming the land areas it initially lost). To strengthen evidence we could present hydrological changes next to conflict data and also discuss counterfactuals, as far as the length of the paper permits.

**References:**

Adams, C., Ide, T., Barnett, J. and Detges, A.: Sampling Bias in Climate–Conflict Research. Nature Climate Change 127, 3: 1–203. https://doi.org/10.1038/s41558-018-0068-2, 2018.

Adger, W.N., Fransen, S., de Campos, R.S., Clark. W.C.: Scientific frontiers on migration and sustainability. 121(3): e2321325121, 2024; DOI:10.1073/pnas.2321325121.

Aquino, G., Guo, W., Wilson, A.: Nonlinear Dynamic Models of Conflict via Multiplexed Interaction Networks. 2019, preprint - arXiv:1909.12457, 2019.

Bukari, K.N., Sow, P., Scheffran, J. (2018) Cooperation and co-existence between farmers and herders in the midst of violent farmer-herder conflicts in Ghana. African Studies Review 1-25.

Dahm, R., Meijer, M., Kuneman, E., van Schaik, L.: What climate? The different meaning of climate indicators in violent conflict studies. Climatic Change 176(11), 2023, DOI: 10.1007/s10584-023-03617-x.

Gioli, G., Hugo, G., Máñez Costa, M., Scheffran, J.: Human Mobility, Climate Adaptation and Development. Introduction to Special Issue, Migration and Development 5 (2): 165–70, 2016; https://doi.org/10.1080/21632324.2015.1096590.

Jin, B., Guo, W.: Data Driven Modeling Social Media Influence using Differential Equations. IEEE/ACM International Conference on Advances in Social Network Analysis and Mining 2022.

Kopp, R.E., Shwom, R.L., Wagner, G., Yuan, J Tipping elements and climate–economic shocks: Pathways toward integrated assessment. 4(8): 346-372. https://doi.org/10.1002/2016EF000362, 2016.

Lenton, T.M., McKay, A., Loriani, S., et al. (eds) The Global Tipping Points Report 2023. University of Exeter, 2023; https://global-tipping-points.org.

Nadiruzzaman, M., Scheffran. J., Shewly, H.J., Kley, S.: Conflict-Sensitive Climate Change Adaptation: A Review. Sustainability 14(13): 8060; https://doi.org/10.3390/su14138060, 2022.

Pham-Duc, B., Sylvestre, F., Papa, F. et al.: The Lake Chad hydrology under current climate change. Scientific Reports 10, 5498, 2020.

Pruitt, J. et al.: Social Tipping Points in Animal Societies. Proc. of Royal Society B, 2018, 285(1887), DOI:https://doi.org/10.1098/rspb.2018.1282.

Scheffran, J., Cannaday, T.: Resistance to climate change policies: the conflict potential of non-fossil energy paths and climate engineering. In: Maas A, et al. (Eds.) Global environmental change: new drivers for resistance, crime and terrorism? (Baden-Baden, Nomos), 2013.

Scheffran, J., Marmer, E., Sow, P.: Migration as a contribution to resilience and innovation in climate adaptation: Social networks and co-development in Northwest Africa. Applied Geography, 33: 119-127, 2012; https://doi.org/10.1016/j.apgeog.2011.10.002.

Vogler, A., Scheffran, J., Schröder, U.: Implications of Russia's Invasion of Ukraine for Decarbonization. In: Engels, A. et al. (Eds.) Hamburg Climate Futures Outlook 2023, Hamburg: Cluster of Excellence Climate, Climatic Change, and Society (CLICCS): 50-51, 2023.

---

## Author Comment (AC2)

**Tipping cascades between conflict and cooperation in climate change**

*Jürgen Scheffran, Weisi Guo, Florian Krampe, Uche Okpara*

**Response to Review Comment (RC2)**

**Summary:** The authors connect research on the dynamics of conflict and cooperation under climate change with research on tipping elements and cascading risks. The scope of the paper fits well with a journal such as ESD, and I agree that there is great potential for modeling and systems dynamics approaches to enrich the study of the climate-conflict nexus. However, after reading it, I cannot recommend this article in its current form for publication since its main intent remained unclear. Is this a literature review (of concepts or model types), a model study, or the application of a model to a case study? Even if the article wants to do everything, the different parts need to be more integrated to justify having them together in a single article.

**Response:** We appreciate the recognition of the great potential of this paper and the call for better integration of the different parts. First we would like to better explain the structure and positioning of the paper before improving integration. After the introductory part on embedding into complexity challenges, the second section is designed as a critical review of the literature on climate and conflict, followed by a review on tipping cascades and models of conflict and cooperation. While there is literature in each of these fields, there is little research connecting them which limits the scope of the reviews. Section 4 makes an attempt to bring them together and develop forward-looking conceptual and integrative approaches which may appear as technical and "hard to follow". The abstract considerations are supplemented and exemplified with the case study of Lake Chad (Section 5). Finally, Section 6 discusses challenges of governance and management of negative and positive tipping in conflict and cooperation, followed by a summary and conclusions. While this may appear as too much or ambitious for one paper, we thought that a pure review paper would be insufficient given the limited literature on the intersection of both fields, without indicating substance on pursuing potential pathways of integration. While this cannot be done in depth here, we still think that it can be beneficial to bring the complementary parts by the coautors into one paper that indicates main avenues and offer a spirit of research that could be pursued later in greater depth and breadth. Following the suggestion, we revise the paper by better integrating the parts, extending some, shortening others or moving them to an Appendix, and clarifying linkages between them in an integrative framework.

**Comment 1:** I wonder about the qualitative difference between "Tipping points and thresholds" and "Risk cascades and chain reactions." Could the latter not be regarded as a single tipping process of a larger, higher-level system?

**Response:** We appreciate this comment, although the difference is not always clear or easy to distinguish because it depends on case-specific circumstances. Each individual system or community can have a tipping point and threshold at which it tips, but as the effects of tipping variables are inducing changes in other variables the question is how far the chain of changes continues and is spreading through the network of connections, leading a single tipping into a tipping cascade affecting the whole system, until a new stable state is reached. Risk cascades and chain reactions induce complex transient behavior that is hard to predict and control. As such we cannot simply say whether the whole system tips or only parts of it.  How far this spreading continues, stops at some point, or even recovers, depends on the degrees of heterogeneity in both space-time and context. Regarding the difference of terms we will refer to other publications (e.g. Lenton et al. 2023; Kopp et al. 2016).

**Comment 2:** It remains unclear what the colour in Fig. 5b represents.

**Response:** The colours are for nodes and graph links. In Figure 5.b, the node colour represents the state of the city, with red meaning conflict and yellow meaning cooperation. The link colour represents relationships, with blue meaning positive and green meaning the lack of any positive evidence (see data and method details in response to Comment 5). A zoomed in graph visualizing the method can be found in Aquino et al. 2019 (page 12), where we can see it in greater detail for eastern Europe and the Levant. We can either cut this part short and associated Figures 5 and 6 to reduce the complexity to key qualitative messages or provide more in-depth explanations and analysis of the underlying model and the data used, as given in the following response to Comments 3 and 4.

**Comment 3:** The results in Figure 6 require a more detailed description of how they were obtained.
**Comment 4:** Figure 6 also lacks label descriptions, which makes it unnecessarily difficult to interpret.

**Combined Response:** Indeed, Figure 6 needs axis labels. In Figure 6a, the 3 axes would be the states or dependent variables of interest (for example: conflict event count, protest count), and what the region of attraction (RoA) analysis shows is that there exists an attractor attracting these states to a single value range. Here, this would correspond to a potential well or a stable equilibrium in Figure 4 (e.g., point x=C according to equation 1).
In Figure 6b, the x-axis is the average loss of supporting graph connections (N) that reduces aid to all graph nodes. As a result, the graph nodes slowly lose performance, sliding from the optimal equilibrium point (C) towards the unstable brink (K), We see each node's x value represented by the lines that are bounded by a theoretical framework. Here, the work by Moutsinas and Guo (2020) uses a random link removal perturbation analysis. As N reduces, a cascade effect occurs, the whole graph reaches a criticality that causes ecosystem collapse.

**Comment 5**: How is the parameter fitting performed when the authors write, "We use historical data to learn the parameters of the model above by fitting independent variables to the dependent variable x."

**Response:** To indicate the possible direction, we refer to the work by one of the coauthors (W.G.), building on a nonlinear dynamic model of conflict via interaction networks (see Aquino et al. 2019 and other sources, cited in Section 4). Here, conflict data $x(t)$ per city/town are used as the node level dependent variables at time $t$, to fit with independent variables that are: historical state of $x(t-1)$ and the weight of graph connections to the node as independent parameters. Equation (2) describes the nonlinear relationship between $x(t)$ and $x(t-1)$, as well as the graph connections with other nodes via the connection matrix $A$. The independent parameters are weighted by the $g(.)$ function: (i) land transport connection ($A$ matrix: 1 or 0), (ii) friendly ties based on existence of economic or political treaties (1 or 0), and (iii) cultural similarity based on religious belief vector of major religions (distance between vectors). We use a multi-variate regression to find the weight of the independent parameters. The data ranges are from 2001 to 2017, and the conflict data (x) is from the Global Terrorism Database (GTD), whereas the trade and transport data are from different UN, CIA and National Geographic databases.

**Comment 6:** It remains unclear whether Figure 7 is purely conceptual or if some model-fitting with empirical data has occurred.

**Response:** Figure 7 is entirely a pictorial narrative on how tipping models can give different tipping dynamics based on how they reinforce or compete with each other through a network. We can make this more clear in the text and better explain.

**Comment 7**: It remains unclear how the model is applied to the case study of Lake Chad. What specific elements of the model go beyond the possibility of having two stable states of cooperation and conflict influence the discussion of the case study? In other words, why is this specific model useful for the case study?

**Response**: Thank you for pointing to this lack of clarity which we hope to remove with more explanations in this response and the manuscript. Within the framing of this paper we largely focus on the duality of conflict and cooperation, as indicated by the paper title. Therefore, we found it useful to demonstrate the applicability of the bi-stable tipping model introduced in Section 4 to the case study of Lake Chad in Section 5, which seems in accordance with RC1. This allows to translate qualitative results from the literature and own research into the modelling frame of tipping points. While we limited ourselves to the two states of conflict and cooperation separated by a tipping point, there can indeed be more than two stable states. Although we wanted to avoid making this problem too complex within the limits of this first paper, we take the comment to make these limits more clear and point to the need for multi-stable approaches in future research.

**Comment 8:** The authors use the passive voice often. As a consequence, I cannot distinguish whether the scientific community is doing something or whether it is the authors who did something. The frequent use of the passive voice makes it difficult for the reader to distinguish the authors' contribution from standard community practices. For example: "Due to the high-dimensional nature of the problem (e.g., number of tipping equation parameters exacerbated by the size of the graph), a range of standard assumptions are often used."

**Response:** We regret if it was difficult to distinguish our work from others. In much of the results sections 4 and 5, in particular in the example case, the work has been carried out by the authors for this paper and our previous work. We will make our contributions more clear in the revision by more active voice, less confusing tense and cited references.

**Comment 9:** Figure 4 could show the parameters C and K.

**Response:** Here, C is the capacity point (highest stable point), and K is the critical tipping point (middle unstable brink) which will be included in Figure 4.

**Comment 10:** In my view, the butterfly effect from chaos theory symbolizes the sensitivity to initial conditions, not necessarily a bifurcation point.

**Response:** Here we rather mean that near a bifurcation or tipping point, small effects can have large consequences. We can adjust wording to make the difference of terms clear.

**Comment 11:** What is the difference between a phase transition and a tipping point or threshold?

**Response:** Phase transition is a broader term of transitions between different states of a system (such as solid and liquid phases of matter) which does not have to be self-enforcing and irreversible tipping from one state without necessarily indicating a new state or reversibility from it, possibly requiring distortional effort.

**Comment 12:** In l. 547, the authors most likely want to refer to Fig. 4, not Fig. 3.

**Response:** Indeed, correct is Fig.4, thank you.

**References**

Aquino, G., Guo, W., Wilson, A.: Nonlinear Dynamic Models of Conflict via Multiplexed Interaction Networks. 2019, preprint - arXiv:1909.12457, 2019.

Kopp, R.E., Shwom, R.L., Wagner, G., Yuan, J Tipping elements and climate–economic shocks: Pathways toward integrated assessment. 4(8): 346-372. https://doi.org/10.1002/2016EF000362, 2016.

Lenton, T.M., McKay, A., Loriani, S., et al. (eds) The Global Tipping Points Report 2023. University of Exeter, 2023; https://global-tipping-points.org.

Moutsinas, G., Guo, W.: Node-Level Resilience Loss in Dynamic Complex Networks. Nature Scientific Reports, 10:(3599), 2020.

---

## Referee Report (RR1)

The authors have addressed all of my comments satisfactorily. I look forward to seeing this in publication.

---

## Author Response (AR2)

**Tipping cascades between conflict and cooperation in climate change**

*Jürgen Scheffran, Weisi Guo, Florian Krampe, Uche Okpara*

**Review comments second round**

**Editor comments:** The reviewers provided considerable additional recommendations for further revising the manuscript. Particularly, please consider the balance between theoretical framework, literature review and Lake Chad case study raised by the reviewers.
* * *
**Report #1: Submitted on 31 Jul 2024 by anonymous referee #3**

Some of the authors are leading figures in the field. This lends them strong credibility within this non-double blind review process. In line with the authors' earlier publications, the article focuses on the crucial effort of utilizing complexity ontology for a more valid policy-guiding climate security nexus research.
However, I see some major issues with the article that should be addressed before seeing the manuscript fit for a more nuanced next revision round.
In particular, it is not always easy to follow the manuscript. I guess that this has several reasons:

1. The manuscript seems to be intended as review article. But it does too many things at once: It reviews existing literature on tipping point modelling and applies tipping point modelling to a case study of Lake Chad. Accordingly, there are passages that serve the one purpose but don't fit well with the other. This applies, for example, to sections 4.1 and 4.2. These are captivating reads for a history-informed review of social science modelling (Seriously, I read it with great interest!). But they are not helpful in preparing the reader for the Lake Chad case study. Given the page limits, it would help to develop this manuscript either into a theory-focused literature review (then drop the empirical case study) or into an empirical case study (then shorten the theory section into a serving section that provides only those aspects that are needed for the empirical part).

**Response:** Thank you for raising questions on context and content of the manuscript and giving advice to make it fit for publication. While the article over large parts is a review article on the linkages between climate conflict, tipping cascades and modeling, it is constrained by the limited and selected number of publications on this intersection and does not aim to review each of the large fields separately (climate, conflict, tipping, cascades, modeling) which would be too much for one review. Based on the review and related concepts we also aim to develop and demonstrate initial ideas and research pathways to integrate these concepts in a modelling framework and case study. At first glance this may appear much for one article but we think the purpose of reviews is not only to repeat what existing publications say, but also to develop ideas for integration and application following from the review, with possible insights for future research. In our view the social science modeling reviewed and described in Sections 4 and 5 introduces methods (such as the Richardson, VIABLE and bi-stable models) relevant to the title and topic of the paper, and are not limited to preparing the Lake Chad case study (Sect. 6 and Fig. 6) which is not meant to be the integrative highlight of the whole paper but an illustration of some of the concepts and methods, in particular the bi-stable model which is also applied in Sect. 5.2 and Fig.5. To address the reviewer's concerns and find a better balance we shortened the manuscript from 32 to slightly above 28 pages and the main text from 13,860 to 11,950 words (incl abstract), in particular in Sections 4.1, 4.2, 5.3 and 7, but keeping the empirical network tipping example in Sect. 5.2 and the Lake Chad case study in Sec. 6 with its model part 6.2 which was explained in more details (including the figure).

2. In the current form, the pursuit of two goals at once also results in a misbalance between an about 16 page-long theory section and an about 6 page-long application section. Within the empirical section it is confusing that further cases (Syria, South Asia) are discussed before the article turns to the actual case. Moreover, chapter 7 discusses "governance challenges" and political options. This is certainly important but follows somewhat isolated.

**Response:** In the previous manuscript we do not count everything until page 16 as theory, much is a review of literature on climate conflict and tipping cascades with references to empirical studies and regional cases ending on page 10. On page 11 we started with the 8-page model Sections 4 and 5 which in the revised paper are reduced to nearly 6 pages. We hope to have now a better balance of the different sections. We do not aim for an indepth case study which here serves as an illustration to the review and theory parts to demonstrate the relevance of the conceptual issues that could be extended in the future. Other cases where climate conflict and tipping are connected (Syria, South Asia) are moved from Sect. 6.1 to the hot spots Sect. 3.4 as a background to the later Lake Chad case, avoiding the streetlight effect. We significantly reduced Sect. 7 on "governance challenges" (together with parts of Sect. 3.5) which we find relevant to address options of positive and negative tipping between conflict and cooperation, which is a core issue of the whole paper and its title.

3. Some passages are not in plain English or difficult to understand. For example: What do the authors mean when they write "Whether climate stress drives a system from undesirable to favourable pathways, from conflict to cooperation, or from vicious or virtuous circles, pathways and opportunities for action consider enabling and constraining conditions."?

**Response:** Sorry if the language was not always clear and adequate to presenting the complex issues. We checked, revised and shortened the whole manuscript in terms of language and understanding, including the mentioned sentence.

4. The article's organization is not always intuitively obvious to the reader. For example, section 6.3 announces that the article will "demonstrate how in the Lake Chad case study meso-scale groups and communities can exhibit diverse and dynamic behaviours when under climate stressors" (line 794-795). Where is this demonstration? Is figure 7 the demonstration? I believe that figures alone are not a sufficient case discussion (adding to the more general point that this article should either develop into a theory-focused literature review or an empirical case study and, in the latter case, expand the empirical part).

**Response:** In addition to streamlining the manuscript, condensing some sections and strengthening connections between the parts, we included a few edits to better explain and integrate the Lake Chad case study. We have enhanced our detailed description relating to Fig. 6 (previously Fig. 7) and how it relates to the Lake Chad narrative, for which evidence of the main pathways is given in Sect. 6.1. In particular, we highlight how internal fragility of societies as well as external forcing by climate stressors, collectively enable transition from cooperation to conflict. We show two cascade pathways: (1) the tipping bias that drought can apply to fishing communities – forcing them away from the lake and lowering the barrier to social transition, and (2) the ensuing power vacuum draws in conflict actors and economies which raises the barrier for transition back to peace even when a wet season returns.

Moreover, a few other aspects:
5. A LOT of passages are insufficiently referenced and need referencing. For several other passages, more recent references exist. Moreover, at least one reference from the main text does not appear in the references list (Kavalski 2015) and another one is spelled differently in the main text and the references (Is it Shaik or Sheik Dahir?)
6. A more content-focused aspect on the impacts from climate change on Lake Chad: What do the authors make of the points raised by Selby and Daoust (10.1080/14650045.2021.2014821, pages 1288-1291) who doubt the claims that Lake Chad is shrinking and that this is due to climate change? Given that Selby and Daoust raise some serious doubts, good references

would be needed to justify the claims (lines 735pp) that lake Chad is indeed shrinking and that this is, if partially, a result of climate change. Moreover, the authors' position on what happens at Lake Chad becomes less clear given that they mention some expansion, rebounding and recovering of the Lake later on (lines 781pp).

**Response to 5:** While the text became shorter, we have included more recent references to an already long publication list, besides removing a few others (including some own work), besides making suggested corrections (Kavalski 2015, Sheik-Dahir). Although there could indeed be many more pulications on each of the individual topics (climate, conflict, tipping, cascades, modeling), we tried to limit the number to those that were instrumental to clarify the linkages and intersections between these fields of research.

**Response to 6**: We do not claim a continued shrinking of Lake Chad and thought we had already taken the doubts about climate-related water levels of Lake Chad into consideration referring to "some expansion, rebounding and recovering of the Lake later on". Our point is that the variability is a challenge which implies that a moving shoreline transversing the Lake's riparian countries is a stressor. To address the concern, we included the following reference and sentence: "Selby and Daoust (2022) find the policy discourse on conflict and security implications of climate change overstated, misleading, and out of line with scientific evidence."
* * *
**Report #2 Submitted on 01 Aug 2024 by anonymous referee #2**

The authors responded to all my comments satisfactorily. I only have two minor observations:

- It remains opaque to me why the authors included the sentence "Going beyond optimal decision-making models" at the beginning of Sect 4.2 and then talked about optimal decision-making in games. Or do the authors argue for considering game theoretic models as agent-based models?

**Response:** We are glad the reviewer is satisfied by our responses. We removed the sentence on "optimal decision-making models" and the connection between game theory and agent-based models.

- It is a pity that the tracked-changes version of the manuscript does not show deletions.

**Response:** Sorry for not showing deleted changes. We assumed that showing the revised parts in blue is sufficient. We will keep tracked changes in this version.
* * *
**Report #3: Submitted on 06 Aug 2024 by anonymous referee #4**

This is a thoughtful, detailed literature review of the approaches to conceptualize and measure tipping points in the Anthropocene (although this term is never mentioned). Such an article is timely and necessary in the current academic, political, and ecological climate and I recommend this paper published with minor revision. Several general comments and specific line edits are outlined in more detail below.

1. There is a concern about decoupling already vulnerable conditions that can easily escalate into a cycle of violence vs. actually assessing the thresholds of trigger points, which you also show expressed concern for on line 247. How to delineate between the two? Is it only places

with the necessary preconditions (i.e. reliance on rainfed agriculture) it is possible to set such thresholds, or any place? How can these be made more context specific?

**Response:** We appreciate the supportive words and included the term Anthropocene. Cycle of violence and thresholds of trigger points are connected as the first can be triggered by the second which is context and place dependent. What matters is the combinations of critical factors for climate change and conflict, and if one is missing the likelihood for the other is lower and may be sub-critical. Rainfed agriculture is a possible connector, being sensitive to climate change while conflict is sensitive to agricultural losses.

There is a conflation of chronic, climate change impacts versus short onset disasters. Surely these would generate differing triggers depending on if it's a short onset hurricane or flood versus a slow onset drought.

**Response:** Time and space matter in tipping processes and have a different scale in short-term events (hurricanes) and long-term events (droughts) which have a different dynamics of tipping and related cascades, where the former require responses on a shorter and smaller time scale while the latter need long-term adjustment over larger regions. A long term disaster acts as a general bias (e.g., an inclination to tip), whereas a short term disaster is more a push/force in one way. There is a short note on page 5

While the literature review is somewhat exhaustive, there seems to be a lack of identifying particular disciplines that focus on AGM. These studies seem to be political science dominated, but what are other disciples doing? Is there a need for interdisciplinary integration? Additionally, the first author's work is cited heavily and major works by other research groups from the climate-conflict literature in political ecology and geography are missing.

**Response:** The literature is meant to be interdisciplinary and not limited to political science. To balance the literature on climate and conflict we reduced those by the first author (who works in geography) and have included references from other authors and fields, including ecology where both system and agent models are popular.

The Lake Chad case study seems like a questionable example given the attention to issues of the 'streetlight' effect but additionally, could be expanded upon considerably. It is not completely clear the temporal scale of the fisherman moving out and leaving a political vacuum and needs considerably more explanation to be a convincing application of your framework. Researchers should be able to apply your VIABLE framework to other case studies.

**Response:** We had mentioned the streetlight effect before and besides Lake Chad shortly discuss climate-conflict cases in other regions (Syria, South Asia) in Sect.3.4. We have now clarified the expectations on what the case study should achieve. It is not meant to be an example of generalizing problems in climate-conflict interaction models, but rather to serve as an illustrative example on how different mechanisms and consequences can be reflected in the model(s) in Sect. 4 and 5. In Sect. 6, we explain: "*Following the climate-security discussion in Sect. 3, the review of tipping models in conflict and cooperation in Sect. 4 and the analysis of the transition dynamics in the bi-stable tipping model of Sect. 5, we now illustrate the conceptions and models for a regional example of a climate security hot spot familiar to many researchers, centred around the Lake Chad region. Translating qualitative narratives to quantitative networked tipping models introduced in Sect. 5, we show how different governance approaches (e.g., support vs. competition) and migration patterns can lead to an erosion or raising of barriers in conflict-cooperation transitions. The purpose is to show how narratives can map to models and not to develop a detailed or generalised Lake Chad climate-conflict scenario which is left to future research, e.g. implementing the VIABLE model.*"

Line edits:

Introduction: A bit reductionist, exaggerates the chaotic aspects of climate change without full discussion of alternative outcomes including peacebuilding and cooperation.

**Response: In** addition to introductory statements "to stabilize the Earth system and to develop forward-looking adaptation policies that prevent violent conflict and enable cooperation" we follow the reviewer by suggesting to "discuss alternative outcomes of positive tipping including peacebuilding and cooperation" which is then further discussed in Sect.3.5 and Sect. 7.

Section 2.2: elaborate more on social tipping points in general- i.e. consider including an example of political tipping points

**Response:** While social tipping points are shortly mentioned in Sec. 2.2 and 3.1 (with few references), examples of social-political tipping points are now given in Sect. 3.4 (hot spots in Syria, South Asia and Lake Chad, further discussed in Sect. 6).

Section 2.3: Turning points in history it could be argued are tipping points? Or are they different ? Be super clear in terminology

**Response:** We replaced "turning point" by "tipping point" to avoid alternative terms.

Section 6.1: Seems this section could be moved up earlier to section 2 to round out the literature review and show the authors are coming from the standpoint of climate change as a "risk multiplier" or mention this point earlier on.

**Response:** We moved other examples from previous Sect. 6.1 (Syria, South Asia and other hot spots) to Sect. 3.4 (not to Sect. 2) where they fit better as part of the literature review on climate change as a "risk multiplier" and hot spots, with reference to the Lake Chad case study in Sec. 6.

---

## Author Response (AR3)

**Tipping cascades between conflict and cooperation in climate change**

*Jürgen Scheffran, Weisi Guo, Florian Krampe, Uche Okpara*

**Final Comments:** EGUSPHERE-2023-1766

**Chief editor decision:** Publish subject to technical corrections.
**Comments to the author**: Both Reviewers and the handling Editor agree that the paper is ready for publication after the minor suggestions indicated by the Reviewers have been implemented.

**Author response:** Thank you for accepting the paper, subject to minor corrections indicated by the Reviewers. Below you find the author response how they are implemented, in addition to correction of typos and adapting the format of references to the guidelines.

**Report #1: Submitted on 13 Feb 2025 by anonymous referee #5**

I think the revised manuscript should be published. It does address various concerns of previous reviewers and makes an important contribution. I have just a couple minor technical/editorial revision suggestions:

**Response:** Thank you very much for the suggested corrections which we have addressed in the manuscript as explained below. Minor corrected typos are not mentioned.

p. 5., line 173, "pre-empted" is the wrong term here, pre-empt means prevent and I don't think that's what the authors mean here, it should be probably "preceded"
p. 5., line 190, "by" should be replaced with "from" I think
p. 5., lines 194-196, the authors write "... where abrupt and extensive climate changes and extreme events could spread through global supply chains..." But climate/weather events don't spread through global supply chains, it's the impacts of these events that can spread, so the sentence needs some rewriting

**Response**: Done as suggested.

p. 6., lines 224-225, sentence "The concept of security..." needs (a) reference(s)

**Response:** A reference for security concepts is: SIPRI, 2022.

p. 8., lines 285, should "These pathways" be replaced with "These risk dynamics". To be honest I find the sentence unclear, it's not clear to me how the listed 5 risks map to the 4 pathways.

**Response:** The wording has been modified in accordance with Fig. 1.

p. 8., lines 314-318, paragraph needs reference(s)

**Response**: Two references included: Eklöw and Krampe, 2019; Rupesinghe and Bøås 2019.

p. 8., lines 320-324, paragraph needs reference(s)

**Response:** Two references incuded: Busby, 2022; Bremberg et al., 2022.

p. 10., line 395, "against" should be replaced by "to", it's "responses to..."
p. 10, lines 402-403, last sentence in the paragraph needs reference(s)

**Response:** One reference is included: Berthet et al., 2024.

p. 11., line 418, what is a "network of sensitivities"? How can tipping cascades spread on a network of sensitivities? This does not make sense to me. I think this needs some rewriting.

**Response:** This part has been rewritten as follows: "Sensitivities between two connected variables measure how a marginal change in one variable affects the other. ... Compounding human responses such as migration, conflict and cooperation, and tipping cascades are spreading through the chain of variables connected by their sensitivities, affecting systemic stability in natural and social systems."

p. 11., line 426, provide reference for "Lanchester Laws"

**Response**: One reference is included: Johnson and MacKay, 2015

p. 12., line 476, I believe there should be a colon after groups and before ABM?
p. 12., I'm not sure what exactly "investments" means in the "VIABLE" model, is that utilisation/ spending of resources, or a costly action? Maybe the authors could add a brief explanation. I guess it does not mean investment in financial terms.

**Response:** Investments mean the capability an agent can apply in the effort to induce system change during action which can be financial resources (e.g. money) or physical resources (e.g. energy). Without going into much detail of the VIABLE model, a short modification is included: "It models the dynamic action and interaction of agents who use part of their available capabilities (K) as efforts (C) invested (such as money or energy)".

p. 13., line 512-513, what does it mean that a system can break apart into simpler ones in response to tipping cascades? Do you mean a society becomes fragmented into smaller social groups?

**Response:** Basically it means fragmenting into smaller social groups, reducing the social connectivity in the interaction matrix. The wording is changed as follows: "In response to the transformation from tipping cascades a system can break apart into simpler ones (a society becomes fragmented into smaller social units with weak connections f in the interaction matrix) or form more complex ones (with stronger interconnections)."

p. 13., Figure 2b, can you please add some more interpretation/explanation of the figure, I find it rather confusing and am not sure how exactly to read it to make sense of it.

**Response:** Figure 2(b) is now explained as follows, including red, green and black colors of the lines: "Lines of satisfaction for two agents where they achieve their target values as a function of both efforts invested. Intersecting equilibria represent mutual satisfaction for red conflicting (--), green cooperative (++), black neutral (0,0) and mixed (+-/-+) relations. Adaptive responses move towards target values and stable equilibria."

p. 14., lines 548-558 add reference(s) in this paragraph?

**Response:** One reference is included: Aquino et al., 2019.

p. 21., line 811-812, "For poor governance community behaviour is facing a low barrier against transition", I'm not sure how to read this, is "poor governance community behaviour" one thing here, or do you mean "Under poor governance, community behaviour is facing..."?

**Response:** The wording is now: "Under poor governance, community behaviour is facing a low barrier against transition".

**Report #2: Submitted on 27 Mar 2025 by anonymous referee #4**
There are still minor spelling errors that should be carefully checked for.

All my comments have been addressed briefly, but satisfactorily. The discussion is more clearly situated in conflict studies and social science more broadly which lends to more powerful arguments of intellectual contributions.

**Response**: Thank you very much for the positive comment. The whole document has been checked once more to hopefully remove all spelling errors. Language format has been made consistent, reference format made compatible with guidelines.